# CONSTRAINED SKILL DISCOVERY: QUADRUPED LOCOMOTION WITH UNSUPERVISED REINFORCEMENT LEARNING

## ABSTRACT

Representation learning and unsupervised skill discovery can allow robots to acquire diverse and reusable behaviors without the need for task-specific rewards. In this work, we use unsupervised reinforcement learning to learn a latent representation by maximizing the mutual information between skills and states subject to a distance constraint. Our method improves upon prior constrained skill discovery methods by replacing the latent transition maximization with a norm-matching objective. This not only results in a much a richer state space coverage compared to baseline methods, but allows the robot to learn more stable and easily controllable locomotive behaviors. We successfully deploy the learned policy on a real ANYmal quadruped robot and demonstrate that the robot can accurately reach arbitrary points of the Cartesian state space in a zero-shot manner, using only an intrinsic skill discovery and standard regularization rewards.

## 1 INTRODUCTION

Learning dynamic motions for legged robots typically requires a combination of complex reward engineering, imitation learning and curriculum learning. This can be limiting in two ways: firstly, reward engineering can be a tedious and time-consuming process, which does not scale well to learning a large range of behaviors; secondly, it constrains the problem through the injection of strong inductive bias. For example, in locomotion it is common to have a reward for moving directly towards the goal — behavior, which can be very sub-optimal if there are any obstructions between the robot and the target. In contrast, skill discovery, or options learning, is a subset of representation learning that presents a promising solution to this problem. It encourages the agent to autonomously explore its environment and learn diverse behaviors with an intrinsic motivation reward (Salge et al., 2013; Choi et al., 2021). The main advantage of this approach is that the agent can learn general behaviors that not only do not require as much handcrafting and reward engineering but can also be easily reused for downstream tasks.

Skills in a latent representation are typically learned by maximizing an information theoretic objective, such as the mutual information between skills and state trajectories (usually with a discriminator) (Eysenbach et al., 2018; Choi et al., 2021), or diversity in terms of expected features and state distributions under different skills (Zahavy et al., 2022; Laskin et al., 2022; Cheng et al., 2024a). However, one of the main limitations of prior methods is that they define mutual information through the Kullback-Leibler (KL) divergence, which does not consider the degree to which different behaviors are distinguishable (Park et al., 2023b). To learn "useful" behaviors —for example state-traversing locomotive behaviors— it is common to constrain the encoder to focus only on certain parts of the observation space, which also injects a strong inductive bias into the problem. Prior work (Park et al., 2021; 2023b) has proposed maximizing the latent transitions in the skill space under a Lipschitz constraint. While such methods can learn skills spanning a great part of the state-space without additional inductive bias, the maximization objective leads to only acquiring overly aggressive high-velocity motions, which cannot be deployed on most robotics hardware.

In this work, we explore the application of skill discovery via representation learning to the domain of quadrupedal locomotion. We introduce an enhanced constrained skill-learning objective that facilitates the robot's acquisition of an extensive repertoire of locomotion skills. Our work builds upon

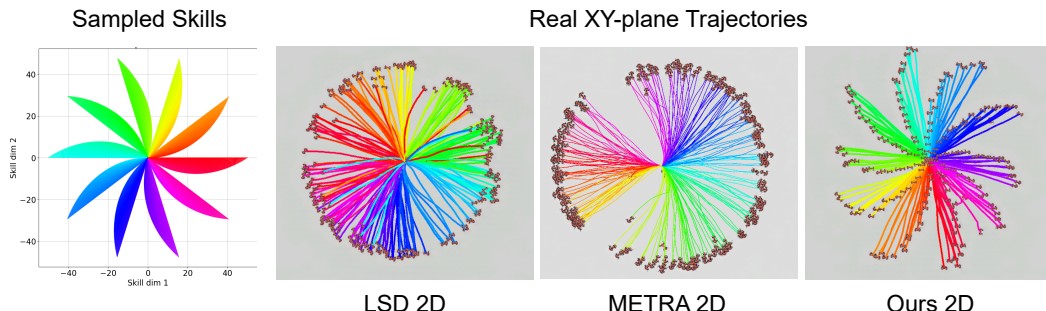

Figure 1: We learn to map skill-conditioned state transitions to a latent space $\mathbb{Z}^2$ through an encoder $\phi(\cdot)$. By training a skill-conditioned policy together with the encoder, we learn a large range of locomotive skills on quadruped robots. Prior methods (Park et al., 2021; 2023b) always maximize the latent transitions, leading to only learning less stable high-velocity motions regardless of the skill magnitude. In contrast, our proposed method learns a wider distribution of behaviors, which we can control by varying the magnitude of the sampled skills.

prior methods for Lipschitz-constrained latent transition maximization Park et al. (2021; 2023b). However, unlike these works, we explicitly avoid always maximizing the latent transition magnitude. The strength of our proposed method is that it learns more stable and controllable skills that cover a wider range of the state space, as shown in Fig. 1. By employing a skill-conditioned policy, our approach enables the robot to navigate the environment at varying velocities. Our method achieves this without using task-related rewards, instead, it relies solely on intrinsic mutual information rewards and extrinsic regularization rewards. Although this skill-conditioned policy facilitates accelerated learning of downstream tasks, we demonstrate that, even without additional training, the policy is sufficiently effective to achieve precise zero-shot goal tracking. Our contributions are as follows:

- We introduce an unsupervised skill discovery approach for pre-training quadruped robots, enabling the acquisition of diverse locomotion skills using only skill discovery and regularization rewards.

- We propose a skill matching objective for constrained skill discovery, enabling the agent to learn a broader range of skills and reliably cover a larger portion of the state space compared to the baseline methods.

- Our method conditions the trained policy on a desired latent space transition, achieving accurate zero-shot goal-tracking on the ANYmal quadruped robot in the real world without the need for additional training.

## 2 PRELIMINARIES AND RELATED WORK

### 2.1 PRELIMINARIES

We formulate a Markov Decision Process (MDP) defined as $\mathcal{M} = \{\mathcal{S}, \mathcal{A}, \mathcal{R}, \mathcal{P}_0(\mathcal{S}), \mathcal{P}(s'|s, a)\}$, where $\mathcal{S}$ and $\mathcal{A}$ are the continuous state and action spaces, respectively, $\mathcal{R}$ is the reward given by the environment, $\mathcal{P}_0(s)$ is the initial state distribution and $\mathcal{P}(s'|s, a) : \mathcal{S} \times \mathcal{A} \to \mathcal{S}$ is the transition probability encoding the dynamics of the environment. For the unsupervised skill discovery problem, we define skills as latent variables $z \in \mathcal{Z}$ in either continuous $z \in \mathbb{R}^N$ or discrete $z \in \{0, 1, \ldots, N\}$ space, with N being the dimensions or the number of skills in the continuous and discrete cases, respectively. We sample a new skill z at the beginning of each episode from a prior distribution $p(z)$ and collect a roll-out trajectory $\tau = (s_0, a_0, \ldots s_T)$ under a state- and skill-conditioned policy $\pi(a|s, z)$. We use Reinforcement Learning (RL) to learn this policy, such that it maximizes the discounted sum of rewards $\mathbb{E}_{\tau \sim p^\pi(\tau)} \left[ \sum_{t=0}^{T} \gamma^t R_t | s_0 = s \right]$ received by the agent from the environment by following the policy, where $\gamma \in (0, 1]$ is a discount factor. The per-step rewards $R_t$ can be a

combination of extrinsic (task-based or regularization) and intrinsic (maximizing the skill diversity) components.

## 2.2 RELATED WORK

**Legged Locomotion:** Reinforcement learning has shown great results in recent years when applied to legged locomotion, both for blind (Hwangbo et al., 2019; Lee et al., 2020; Kumar et al., 2021; Margolis et al., 2024; Yu et al., 2023; Gangapurwala et al., 2023) and for perceptive locomotion (Miki et al., 2022; Loquercio et al., 2022; Yang et al., 2023; Gangapurwala et al., 2022). The standard approach is to frame the learning problem as a velocity-following task, where the robot learns to track user-specified velocity commands under noise and various disturbances. One disadvantage of this method is that the policy often converges to a single gait behavior, unless additional rewards (Fu et al., 2022) or neural network structures (Yang et al., 2020) are included. Recently, more works have explored learning navigation and locomotion, resulting in a goal-conditioned policy that can reach various points in the Cartesian state space while allowing the robot to adapt its velocity depending on the environment. Due to the sparsity of the task rewards, many works implement additional dense exploratory rewards (Rudin et al., 2022a; Zhang et al., 2024; Cheng et al., 2024a), or task-specific curricula (Li et al., 2024; Atanassov et al., 2024). Cheng et al. (2024b), for example, use a simpler velocity-based reward but requires pre-specified oracle waypoints for the robot to follow as it learns to navigate more challenging environments. Other parkour works (Zhuang et al., 2023; Hoeller et al., 2023) require learning several manually defined skills, each with its own set of task-based rewards that require separate design and tuning. To summarize, legged locomotion by reinforcement learning needs heavy tuning of task-specific rewards, which requires domain knowledge and human efforts.

**Unsupervised Skill Discovery:** To address the challenges of sparse reward problems that require strong and structured exploration, many works have proposed encouraging curious and exploratory behavior through intrinsic motivation. There are many methods for representing intrinsic motivation in reinforcement learning agents, which can be broadly grouped into three information-theoretic objectives - maximizing expected information gain, novelty, and empowerment (Salge et al., 2013; Jung et al., 2012) (also referred to as options learning (Sutton et al., 1999; Bacon et al., 2016; Jong et al., 2008) and skill discovery) (Aubret et al., 2023). Of these three methods, skill discovery is particularly attractive as it promises to encourage exploration throughout training and to learn a reusable representation or set of behaviors for downstream tasks. Learning state-traversing behaviors has also been done through goal-conditioned RL (GCRL) (Kaelbling, 1993; Pong et al., 2020; Durugkar et al., 2021; Cho et al., 2023), and in fact Choi et al. (2021) have linked these two ideas together under the name of variational GCRL. Some approaches (Bacon et al., 2016; Harb et al., 2017) learn both the skills and a high-level skill selector jointly. However, as discussed in (Eysenbach et al., 2018), this can lead to a modal collapse where suboptimal options are picked less often, and as such, not improved further. Many skill discovery methods then approach this separately - by first learning skills, and then training a high-level meta controller which executed skills for a certain number of time steps. Some options frameworks (Bacon et al., 2016; Harb et al., 2017; Li et al., 2022) learn termination functions for each skill instead.

To learn diverse behaviors, prior works commonly use the mutual information (MI) between skills and states, $I(\mathbf{z}; \mathbf{s})$ as the optimization objective (Choi et al., 2021). Intuitively, this encourages the agent to learn to associate different states (or trajectories of states, i.e. behaviors) with different skills. The MI can be expressed in two ways - the first being in terms of the skill-conditioned state entropy and the entropy of the states, i.e. $I(\mathbf{z}; \mathbf{s}) = H(\mathbf{s}) - H(\mathbf{s}|\mathbf{z})$ (Liu & Abbeel, 2021; Sharma et al., 2020a;b; Laskin et al., 2022); or alternatively as $I(\mathbf{z}; \mathbf{s}) = H(\mathbf{z}) - H(\mathbf{z}|\mathbf{s})$, where the skill entropy $H(\mathbf{z})$ depends on the choice of distribution $\mathbf{z} \sim p(\mathbf{z})$, typically chosen as uniform and is therefore constant. As maximizing $H(\mathbf{z}|\mathbf{s})$ directly is intractable, a variational approximation is used with the following lower bound:

$$I(\mathbf{z}; \mathbf{s}) = H(\mathbf{z}) - H(\mathbf{z}|\mathbf{s}) \tag{1}$$

$$= -\mathbb{E}_{\mathbf{z}}[\log p(\mathbf{z})] + \mathbb{E}_{\mathbf{s},\mathbf{z}}[\log p(\mathbf{z}|\mathbf{s})] \tag{2}$$

$$\geq -\mathbb{E}_{\mathbf{z}}[\log p(\mathbf{z})] + \mathbb{E}_{\mathbf{s},\mathbf{z}}[\log q_\phi(\mathbf{z}|\mathbf{s})] \tag{3}$$

$$\geq c + \mathbb{E}_{\mathbf{s},\mathbf{z}}[\log q_\phi(\mathbf{z}|\mathbf{s})], \tag{4}$$

where $q_\phi(\mathbf{z}|\mathbf{s})$ is a variational approximation of the conditional pdf $p(\mathbf{z}|\mathbf{s})$. This can be approximated with a discriminator for both discrete (Gregor et al., 2017; Eysenbach et al., 2018; Achiam et al.,

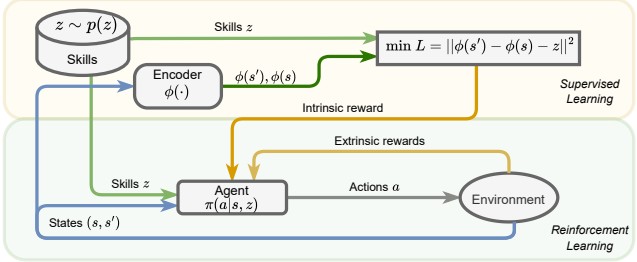

Figure 2: Learning scheme for our proposed approach. The encoder $\phi(\cdot)$ maps state transitions into a latent space optimized to match the skills (sampled from a predefined distribution $p(z)$), as shown by the MSE loss. An intrinsic reward is given to the agent based on the loss magnitude and an extrinsic reward from the environment, which encourages smooth behaviors.

2018) and continuous (Choi et al., 2021; Hansen et al., 2020) skill spaces, and used as the reward function for the agent $\pi(a|s, z)$ in an RL setting. This works as a cooperative game, where both the agent's and discriminator's objectives are to maximize the lower bound. In VALOR (Achiam et al., 2018) the complete episodic trajectory is used to distinguish states, but it is more common to use combinations of current, next, initial or final states to simplify the problem (Gregor et al., 2017; Park et al., 2021). In DISDAIN (Strouse et al., 2021) the reward is augmented to use the disagreement between an ensemble of discriminators, which helps address the problem of the discriminator making incorrect predictions on novel states it hasn't encountered. The same intrinsic motivation formulation is also used in CASSI (Li et al., 2023) in combination with adversarial motion imitation (Peng et al., 2021) to learn discrete diverse skills from a motion dataset. Similarly, ASE (Peng et al., 2022) uses an encoder-based representation to learn continuous high-dimensional skills given a motion dataset, but constraints the latent space to lie on the surface of an N-dimensional hypersphere. One major disadvantage of the common formulations is the lack of exploration incentive (Laskin et al., 2022). In fact, as long as the discriminator can distinguish between skills there is no benefit to exploring more diverse states (Park et al., 2021). With a large enough discriminator, even a small difference in the states could lead to accurate skill predictions. This causes many unsupervised skill discovery methods to produce distinguishable skill-conditioned trajectories that cover a reasonably small part of the state-space. To tackle this, for locomotion it is common to add a high level of task inductive bias into the problem, for example by limiting the discriminator input to the XY position of the body (Eysenbach et al., 2018; Sharma et al., 2020b). Alternatively, giving structure the problem by imitating a reference motion dataset (Peng et al., 2022; Li et al., 2023) or an optimal pretrained policy (Zahavy et al., 2022; Cheng et al., 2024a) can also significantly help with converging to more diverse behaviors. Recently, Park et al. (2021) proposed an alternative formulation to align the transitions in a learned latent space with the corresponding skills and maximize their norm. Crucially, they then add a constraint on the norm of these latent transitions being a lower bound on a chosen distance metric, such as the Euclidean (LSD) (Park et al., 2021) or the temporal distance (METRA) (Park et al., 2023b) between a transition pair in the original state space. This constraint ensures that skills that cover large latent distances also cover large state space distances under the chosen metric, leading to much larger coverage and more useful skills.

Unsupervised skill discovery has mainly been studied in relation to standard problems (such as the MuJoCo robotic environments) in the domain of reinforcement learning. Few skill discovery methods have been applied to robotics and tested in the real world. Sharma et al. (2020a) applied an off-policy version of DADS (Sharma et al., 2020b) for real-world reinforcement learning on a quadruped robot. Recently, Cheng et al. (2024a) applied DOMiNO (Zahavy et al., 2022) to goal-conditioned quadrupedal locomotion, where the discrete skills are used to produce diverse trajectories that reach the same goal point. However, their approach requires an optimal pretrained policy, that is already capable of reaching the goal.

## 3 METHODOLOGY

Our approach builds upon the method proposed in (Park et al., 2021; 2023b), where we learn an encoder $\phi$ that maps state transition pairs to a latent space. As shown in Fig. 2, we sample a 2-D

continuous skill vector and optimize both the encoder and the skill-conditioned policy $\pi(\mathbf{a}|\mathbf{s}, \mathbf{z})$ with the common objective of minimizing the prediction error between latent embedding and the sampled skill.

## 3.1 SKILL DISCOVERY

In (Park et al., 2021; 2023b) the authors show that by parameterizing the discriminator in (4) with a normal distribution with mean $\phi(\mathbf{s})$ and unit variance, the mutual information objective can be reformulated as follows:

$$I(\mathbf{z}; \mathbf{s}) \geq \mathbb{E}_{\mathbf{s},\mathbf{z}}[\log q_\phi(\mathbf{z}|\mathbf{s})] + c = -\frac{1}{2}\mathbb{E}_{\mathbf{s},\mathbf{z}}[||\phi(\mathbf{s}) - \mathbf{z}||^2] + c \tag{5}$$

The initial $\mathbf{s}_0$ and final states $\mathbf{s}_T$ of the episode are used as input to the latent space encoder and $\phi(\cdot)$ with the following decomposition for the loss:

$$\mathcal{L} = -\frac{1}{2}\mathbb{E}_{\mathbf{s},\mathbf{z}}[||(\phi(\mathbf{s}_T) - \phi(\mathbf{s}_0)) - \mathbf{z}||^2] + c \tag{6}$$

$$= \underbrace{-\frac{1}{2}\mathbb{E}_{\mathbf{s},\mathbf{z}}[(\phi(\mathbf{s}_T) - \phi(\mathbf{s}_0))^T(\phi(\mathbf{s}_T) - \phi(\mathbf{s}_0))]}_{\text{L2 Regularization}} + \underbrace{\mathbb{E}_{\mathbf{s},\mathbf{z}}[(\phi(\mathbf{s}_T) - \phi(\mathbf{s}_0))^T\mathbf{z}]}_{\text{Directional Alignment}} - \underbrace{\frac{1}{2}\mathbb{E}_{\mathbf{s},\mathbf{z}}[\mathbf{z}^T\mathbf{z}]}_{\text{Constants}} + c$$

$$\tag{7}$$

LSD (Park et al., 2021) and METRA (Park et al., 2023b) then make the following simplification to the objective:

$$\mathcal{L}^{LSD} = \underbrace{\mathbb{E}_{\mathbf{s},\mathbf{z}}[(\phi(\mathbf{s}_T) - \phi(\mathbf{s}_0))^T\mathbf{z}]}_{\text{Directional Alignment}}, \tag{8}$$

where the squared skill term $\mathbb{E}_{\mathbf{s},\mathbf{z}}[\mathbf{z}^T\mathbf{z}]$ can be ignored as it is constant (given a fixed skill sampling distribution). The effect of this simplified decomposition, which ignores the squared $\phi$ regularization term, is that the latent space transition $\phi(\mathbf{s}_T) - \phi(\mathbf{s}_0)$ should be aligned with the direction of the skill vector $\mathbf{z}$, and maximized. By combining this optimization objective with a constraint on the latent space transition norm of the form $\forall \mathbf{s}_T, \mathbf{s}_0 \in \mathcal{S}, \quad ||(\phi(\mathbf{s}_T) - \phi(\mathbf{s}_0))|| \leq d(\mathbf{s}_T, \mathbf{s}_0)$, their method prioritizes learning skills which maximize the corresponding distance (according to the chosen metric $d$) between initial and final states.

The drawback of the original LSD and METRA formulations is that the agent will always maximize the norm of the latent space transition (and thus the distance under the metric $d$). In a locomotion scenario, with Euclidean distance as the metric as in (Park et al., 2021), this would result in the agent accelerating as fast as possible in a different direction on the XY plane. This means that while the policy would cover a large part of the state space, most of these states would be transient - i.e. to reach the distant states, the agent has to pass through the near states, but it will not stay there. Staying within the locomotion scenario, this would translate to all skills having a large mean linear velocity, with no low velocity skills. This poses a significant problem if we want to deploy these motions on robot systems. In addition, these slower skills might have been necessary for downstream tasks, causing the performance to deteriorate. Therefore, we propose an alternative objective function while maintaining the same distance metric constraint as in (Park et al., 2021; 2023b). Instead of only optimizing for the alignment between skills and latent trajectories, we also include norm matching. If we use a Mean Squared Error loss, we can recover the original more general formulation in (6), which can be seen as a continuous regressive discriminator form of DIAYN (Eysenbach et al., 2018; Choi et al., 2021). This results in the following objective:

$$J^{\text{Ours}} = -\frac{1}{2}\mathbb{E}_{\mathbf{s},\mathbf{z}}[||(\phi(\mathbf{s}_T) - \phi(\mathbf{s}_0)) - \mathbf{z}||^2], \quad s.t. \quad ||\phi(\mathbf{s}_T) - \phi(\mathbf{s}_0)|| \leq ||\mathbf{s}_T - \mathbf{s}_0|| \tag{9}$$

To make the objective suitable for solving as an RL problem where we want to optimize the objective from a batch of environmental transitions, we define the per-step loss similarly to (Park et al., 2021):

$$J_t^{\text{Ours}} = -\frac{1}{2}\mathbb{E}_{\mathbf{s},\mathbf{z}}[||N \cdot (\phi(\mathbf{s}_{t+1}) - \phi(\mathbf{s}_t)) - \mathbf{z}||^2], \quad s.t. \quad ||\phi(\mathbf{s}_{t+1}) - \phi(\mathbf{s}_t)|| \leq ||\mathbf{s}_{t+1} - \mathbf{s}_t||, \tag{10}$$

where $\mathbf{s}_t$ and $\mathbf{s}_{t+1}$ are a pair of consecutive states, and N is the number of steps in the episode. While this telescopic sum is not equal to the original formulation, as in (Park et al., 2021) (due to the use of the squared norm), we show in the Supplementary Materials that it is an upper bound.

To then encourage the skill-conditioned policy $\pi(\mathbf{a}|\mathbf{s}, \mathbf{z})$ to produce diverse motions, we give it a reward based on how accurate the discriminator is:

$$r(t) = \frac{1}{1 + \sigma e}, \quad \text{where} \quad e = ||N \cdot (\phi(\mathbf{s}_{t+1}) - \phi(\mathbf{s}_t)) - \mathbf{z}||^2, \tag{11}$$

where $\sigma$ is a scaling term to account for the magnitude scale of the loss. An alternative way to interpret this objective is that $\phi(\cdot)$ is an encoder that compresses states into a latent space, and our objective pushes the agent to traverse this latent space in the same direction and magnitude as the skill it is conditioned on. By constraining this latent space transition with the state space Euclidean norm, we can ensure that we cover a distance of at least that magnitude in the original state space. Unfortunately, this upper bound does not guarantee that the real traveled distance will be of similar magnitude to the latent one. In fact, all skills could traverse a very large real space distance and still satisfy the constraint in (9). However, in practice, this was not an issue as the agent would prefer stationary behavior over highly dynamic motions due to its extrinsic regularization rewards.

We train the encoder $\phi(\cdot)$ using supervised learning using the objective in (10) and the RL policy using the intrinsic reward in (11), on state pairs from the environment and the skills they were produced under. This can be seen as a cooperative game, where the agent tries to produce skill-conditioned trajectories that are more easily distinguishable by the discriminator. Since the L2 loss can be more sensitive to large-scale values, we use the Smooth L1 loss instead for training the encoder $\phi(\cdot)$. While we use a Euclidean distance constraint (Eq. 10), in Appendix A.7 we show that our approach can also work well with other types of distance metrics, such as temporal distance.

## 3.2 REINFORCEMENT LEARNING SETUP

**Observations, actions and control framework.** For the policy observations, we use a concatenated array of current joint positions and velocities, base linear and angular velocities, base quaternion and the previous actions. While some previous skill discovery works (Eysenbach et al., 2018; Sharma et al., 2020b) require constraining the discriminator input based on the task (e.g. to only the XY base position for locomotion tasks), this is not the case for our method, similarly to (Park et al., 2021). For this reason, we use the full observations, i.e. the base linear $\mathbf{v}_b$ and angular $\boldsymbol{\omega}_b$ velocities, base quaternion $\bar{\mathbf{q}}_b$, joint positions $\mathbf{q}_j$ and velocities $\dot{\mathbf{q}}_j$, and the base position $\mathbf{p}_b$. The action space consists of desired joint positions for the 12 joints on the ANYmal robot relative to a nominal standing configuration. These are updated at $50\,\text{Hz}$ and summed up with the nominal joint positions and passed at $400\,\text{Hz}$ to low-level higher frequency PD controllers.

**Reward design.** To slightly bias the skill discovery towards feasible motions that can be applied to the real robot without damaging the hardware, we provide several extrinsic rewards commonly used in legged locomotion. These are energy conservation and smoothness terms, together with feet air time (to prevent high-frequency stepping), flat orientation and nominal base height rewards, and unwanted contact penalties. We note that these extrinsic rewards are only necessary to make the motions smoother and more aesthetically pleasant. In the supplementary material we conduct an ablation with and without those extrinsic regularization rewards, and show that they are not crucial for learning the diverse behaviors. In comparison to quadruped locomotion acquired by pure reinforcement learning, we use far fewer reward terms since we don't need any task-specific rewards.

## 4 EXPERIMENTAL RESULTS

We demonstrate the ability of our approach to discover a broad range of skills which are used to locomote to desired goal poses. These learned skills are used to achieve zero-shot goal-tracking in the real world. We compare our method with two state-of-the-art baselines: (i) **Lipschitz-constrained skill discovery (LSD)** (Park et al., 2021) with the objective shown in (8) and a Euclidean distance constraint; (ii) **Metric-aware abstraction (METRA)** (Park et al., 2023b) with the same objective as LSD and a temporal distance constraint. As in the baselines, we choose a 2D skill space, and therefore a 2D latent space for the encoder $\phi(\cdot)$, and sample the skills uniformly from within a circle of radius $||z_{\max}|| = 50$. We choose this value because for a 300 step episode it results in a per-step latent transition of $\sim 0.17$, which is the maximum magnitude achieved by LSD on the same task. For the rest of the experimental setup, please refer to the supplementary material.

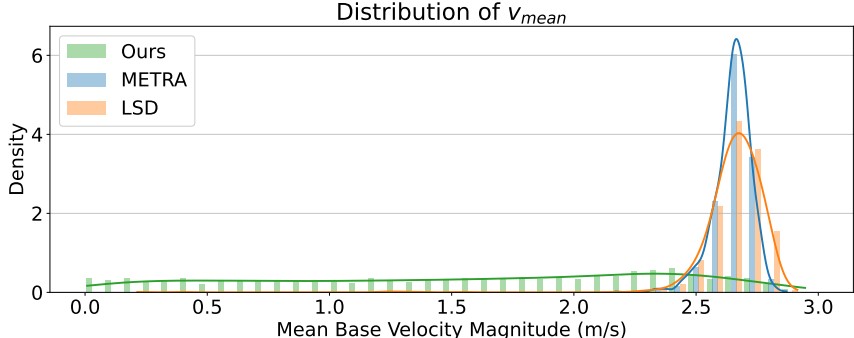

Figure 3: Density distribution of the mean (across the episode) base velocity of 1000 trajectories with uniformly sampled skills, grouped into equally spaced bins in the range $0\,\mathrm{m/s}$ to $3\,\mathrm{m/s}$. We show the results for the baseline LSD (in orange), METRA (in blue), and ours (in green). A broad distribution is a result of a larger skill space.

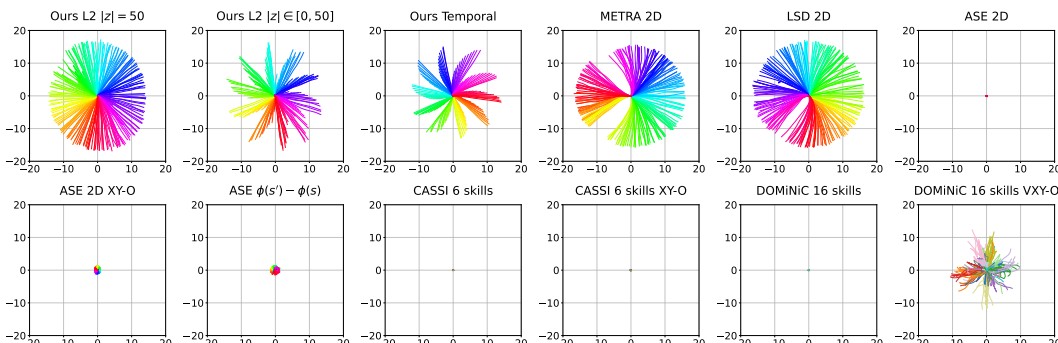

Figure 4: Comparison of XY base position trajectory (in meters) between ours, METRA, LSD, ASE, CASSI and DOMiNiC. To better illustrate the magnitudes of difference in the performance, we show the results with a fixed x- and y-axis scale across all algorithms. The colors indicate different skills (whether discrete or continuous). For ours, we show the performance when sampling skills with both the maximum magnitude, and with varying magnitudes in the first two plots, respectively. All methods use the full observation space, with the exception of those variations annotated as "XY-O" and "VXY-O", which only contain base position (or velocity, respectively) as the observation.

### 4.1 STATE SPACE COVERAGE

We compare our approach to LSD and METRA in terms of the mean base velocity magnitude across the skill space, as illustrated in Fig. 3. Both LSD and METRA exhibit a very narrow distributional range of base velocities, with almost all skills exhibiting high velocities. In contrast, our method can achieve a much more uniform spread, which enables the learning of both low- and high-speed locomotive behaviors.

### 4.2 SKILL DISCOVERY COMPARISON WITH BASELINE METHODS.

In this section, we compare the performance in terms of XY state coverage with existing skill discovery methods - namely METRA (Park et al., 2023b), LSD (Park et al., 2021), CASSI (Li et al., 2023), ASE (Peng et al., 2022) and DOMiNiC (Cheng et al., 2024a). These methods have been shown to learn impressive diverse skills on both quadruped robots and physically simulated characters. For a fair comparison, we evaluate purely the skill discovery parts of the methods **without** the motion imitation dataset and objective (or the optimal pretrained policy in the case of DOMiNiC). For our methods, we compare the performance with a Euclidean (L2) and a Temporal distance constraint (for details please refer to Appendix A.7). To show that we learn a wider range of behaviors, for the Euclidean variation we show the performance at inference time using skills with max magnitude ($|z| = 50$) and varying magnitude ($|z| \in [0, 50]$).

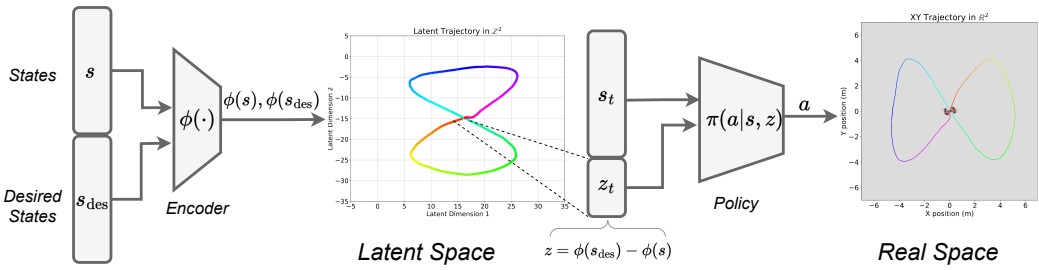

Figure 5: After training, we can plan in the learned latent representation to condition the policy to reach desired states. We encode the current state $\mathbf{s}$ and desired state $\mathbf{s}_{\mathrm{des}}$ into the latent space, and use that as the conditioning skill for the policy.

As can be seen from Fig. 4 both LSD and METRA learn a simpler latent representation where the same latent (and real) transitions are achieved regardless of the magnitude of the sampled skill - and the magnitude of these transitions is always explicitly maximized. On the other hand, our method produces both low- and high magnitude transitions, depending on the sampled skill. 2-D ASE results in behaviors that do not traverse the XY state space, but only affect the joint positions of the robot. By constraining the encoder input to only the base XY position (ASE 2D XY-O), or increasing the dimensionality of the skill space, the robot learns to locomote across the XY plane. However, compared to our method, ASE exhibits much lower state space coverage. The reason for this is that the minimization of the encoder loss only ensures diverse behaviors, but not the degree to which they differ. Similar observations were noted in (Park et al., 2021) when comparing to other methods like DIAYN and DADS. Even when constraining the encoder to only look at the XY position, it converges to small movements that are distinguishable but not useful for traversing the state space. We note that ASE uses almost the same objective as LSD and METRA — but with a von Mises-Fischer distribution rather than a Gaussian, a slightly modified loss ($\mathcal{L} = \phi(s, s')^T z$ rather than $\mathcal{L} = (\phi(s') - \phi(s))^T z$), and crucially, without the constraint on the latent distance seen in Eq. 10. For fairness, we evaluate both types of losses (where the former are labeled with $\phi(s') - \phi(s)$). We believe the lack of maximization of the latent transitions and lack of constraint between the real states and the latent ones results in the subpar performance of ASE compared to ours, LSD and METRA.

Similarly, when using the full observation as input to the discriminator, CASSI mostly relies on the joint position as a distinguishing feature for different skills. If the input is constrained to the XY position (CASSI 6 skills XY-O), different skills result in small XY state space traversal. However, as in ASE, since there is no incentive to maximize this traversal, each skill only covers a small part of the space. Our version of DOMiNiC does not normalize the successor features and thus the objective can maximize them to learn more diverse skills. With either the full observation or just the XY position as discriminator inputs (the latter not shown here for conciseness), DOMiNiC is not able to traverse the XY state space. However, when constraining the input to the base linear velocity only ("VX-O"), the method can discover locomotive behaviors. The state space coverage is smaller compared to ours and the skills are discrete, which limits their potential scalability and usefulness for downstream tasks.

### 4.3 ZERO-SHOT GOAL-TRACKING

Similar to (Park et al., 2021), our method can also achieve zero-shot goal-tracking after training the skill-conditioned policy. As the latent space transition is aligned with its corresponding skill, we can invert the problem and select a skill based on the desired latent space transition, as $\mathbf{z}_{\mathrm{des}} = \phi(\mathbf{s}_{\mathrm{des}}) - \phi(\mathbf{s}_t)$, where $\mathbf{s}_{\mathrm{des}}$ and $s_t$ are the desired final state and the current state, respectively. We set $\mathbf{s}_{\mathrm{des}} = \mathbf{s}$ for all states except the base position, which is set to the desired goal. By conditioning the policy on the desired skill $\pi(\mathbf{s}_t, \mathbf{z}_{\mathrm{des}})$, the robot will produce a motion that reaches $\phi(\mathbf{s}_{\mathrm{des}})$ from $\phi(\mathbf{s}_t)$ in $t_{ep}$ seconds, which is the duration of the episode throughout training. However, in order to ensure that the robot can stop at the desired state, we update the direction and magnitude of the desired skill $\mathbf{z}_{\mathrm{des}}$ at every step (in a closed loop). In contrast, if we keep $\mathbf{z}_{\mathrm{des}}$ constant and do not update it, the robot would move in the direction of the goal with a constant velocity, cross the goal

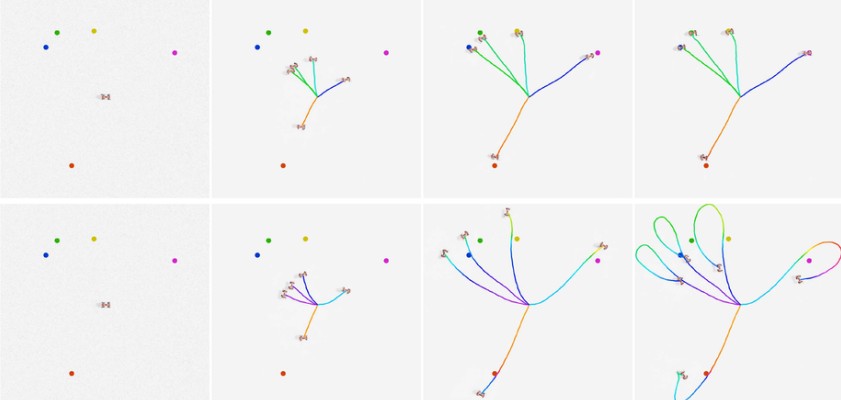

Figure 6: Goal-tracking comparison between Ours (top) and the baseline LSD (bottom). As our skill space can exhibit a larger range of velocities, it can accurately reduce its velocity, and reach and stay at the target. The color gradient on the trajectories indicates the change in skill direction.

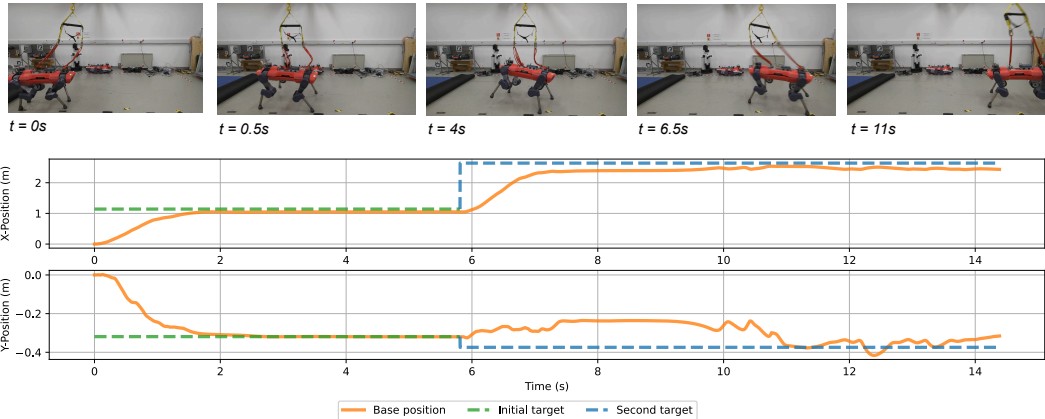

Figure 7: Position tracking performance and corresponding body position profile in hardware experiments. The robot is able to accurately track two successive target poses.

and keep moving along the same direction. As shown in Fig. 6, the former results in highly accurate goal-tracking behavior without additional training. In comparison, while the baseline method moves in the direction of the goal, most of its skill space is dedicated to high velocity motions, causing the robot to repeatedly overshoot the target. We further validated our approach on the real ANYmal robot, as shown in Fig. 7, where the robot has to track two consecutive targets. As can be seen, the robot can accurately reach and stop at the desired goal.

## 4.4 GOAL REACHING IN HIGHER DIMENSIONAL (3-D) SKILL SPACES

The encoder learns to compress its input into the latent space in a way that preserves both the magnitude and the direction of the corresponding skills. In this case, this results in mapping the base linear velocities and the base position and ignoring the remaining dimensions of the input states. The reason for this is that the encoder has a limited latent space dimensionality and those are the dimensions that can achieve the largest transition magnitudes. An interesting hypothesis is that by increasing the latent dimensionality, the encoder should use a larger part of its input state space, and subsequently be able to perform zero-shot goal tracking on the rest of its state space. To investigate this, we trained our approach with a 3-D latent (and skill) space. In Fig. 8 we condition the policy on tracking different base positions with an arbitrary desired yaw. As can be seen in the figure, the agent can successfully move to the desired point with the required heading simultaneously, which was not possible with 2-D skills.

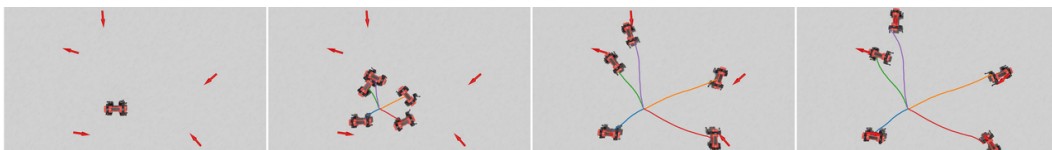

Figure 8: Tracking the desired position with desired orientation (shown by the direction of the red arrow), starting from the origin with a 3-D latent space encoder.

## 5 CONCLUSION

In this work, we presented a novel approach that uses unsupervised skill discovery to learn robust locomotion skills. We showed that these cover a much larger part of the state space when contrasted to baseline methods. We introduced an augmented mutual information objective function, so that an encoder learns to map input states to a latent space and predict both the direction and magnitude of the sampled skills. By combining this with a constraint on the Euclidean distance between states, as in previous work, our method resulted in a much richer state space coverage. In addition, we show that our approach can scale well beyond Euclidean norm to other distance metrics, such as temporal distance. Furthermore, we demonstrate that it can learn useful behaviors to navigate complex environments, such as a 2-D maze. While these skills can be used for various downstream tasks, in this work, we showed how the robot can accurately reach desired Cartesian positions in a zero-shot manner. Our method achieves this without any task-relevant rewards, unlike other works which require carefully engineered dense exploratory rewards. Compared to the baselines (Park et al., 2021; 2023b) we achieve a much more uniform skill-conditioned state distribution. In contrast with other intrinsic motivation works that require an imitation dataset (Peng et al., 2022; Li et al., 2023) or a pretrained optimal policy (Zahavy et al., 2022; Cheng et al., 2024a), our method can learn a diverse set of motions with very little prior structure.

**Limitations and future work.** In this work we used up to 3-D continuous skill spaces, which allow for control of both position and orientation of the robot. However, since our formulation is inherently more challenging to learn than the LSD and METRA baselines, as we learn both magnitude and direction, the encoder is more difficult to train for larger skill dimensions. A possible explanation is that the encoder input does not contain enough information to predict high dimensional skills. For example, some states like the pitch and roll cannot be changed much without resulting in the robot falling; other states like those describing the joint and base motion are co-dependent and intertwined for complex articulated systems like quadruped robots. Nevertheless, in future work, we will investigate whether more complex model structures can be leveraged to train accurately for even higher continuous skill dimensions.

An interesting research direction would be to evaluate the performance at more downstream tasks than just goal-tracking. Exploring the application of unsupervised skill discovery methods to loco-manipulation tasks is an exciting possibility. Prior work has shown that such methods can learn useful skills in robot manipulator setups, and investigating the kind of behaviors the agent acquires for simultaneous locomotion and manipulation is a promising research direction.

**Reproducibility Statement:** To contribute towards reproducible research, we have included the hyperparameters and rewards necessary to reproduce our work. Our RL set-up is based on the SKRL (Serrano-Muñoz et al., 2023) library and the environments are based on the Isaac Sim Orbit framework (Mittal et al., 2023). We will make our code base publicly accessible upon acceptance.

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

| Hyperparameter | Value |
|---|---|
| PPO clip ratio | 0.2 |
| Value clip ratio | 0.2 |
| Entropy Loss scale | 0.1 |
| Value loss scale | 1.0 |
| Optimizer | Adam |
| Learning rate | Adaptive |
| KL divergence threshold | 8.e-3 |
| Discount Factor $\gamma$ | 0.99 |
| GAE Lambda | 0.95 |
| Rollouts per update | 24 |
| Learning epochs | 15 |
| Mini-batches | 4 |

Table 1: PPO hyperparameters.

# A  APPENDIX

## A.1  PER STEP OBJECTIVE LOSS

$$\min \mathcal{L} = ||(\phi(\mathbf{s}_T) - \phi(\mathbf{s}_0)) - \mathbf{z}||^2 \tag{12}$$

$$= ||\sum_{t=0}^{T}(\phi(\mathbf{s}_{t+1}) - \phi(\mathbf{s}_t) - \frac{\mathbf{z}}{T})||^2 \tag{13}$$

$$\leq \sum_{t=0}^{T} ||(\phi(\mathbf{s}_{t+1}) - \phi(\mathbf{s}_t) - \frac{\mathbf{z}}{T})||^2, \tag{14}$$

where T is the number of steps in the episode and (14) follows from the triangle inequality. Since the RHS in (14) is an upper bound to the minimization objective, we can minimize the per step loss at every step of the RL algorithm and ensure the true loss is also minimized.

## A.2  TRAINING SETUP

For the encoder, we use a simple 3-layer [256,128,64] MLP with ReLU activations, and for the policy, we use a 3-layer [512,256,128] MLP with ELU activations. We train both models for 100k environmental steps in the Isaac Sim simulator using the Orbit framework (Makoviychuk et al., 2021; Mittal et al., 2023). For the policy we use Proximal Policy Optimization (PPO) (Schulman et al., 2017) with hyperparameters reported in Tab. 1. The encoder is trained in a supervised manner, using Smooth L1 loss and the Adam optimizer, as shown in Tab. 2.

## A.3  HARDWARE SETUP

To perform zero-shot goal tracking, we require the base position as an observation ot the encoder. On the hardware, we use the manufacturer's ANYmal robot state estimator which directly provides estimates of the base position in the world frame, relative to the initial position of the robot.

## A.4  REWARD TERMS

Table 3 shows the extrinsic rewards we use in our problem (based on those used in (Rudin et al., 2022b)).

| Hyperparameter | Value |
|---|---|
| Lagrange multiplier | 30. |
| Lagrange multiplier learning rate | 1.e-4 |
| Lagrange multiplier slack | 1.e-6 |
| Optimizer | Adam |
| Encoder learning rate | 5.e-3 |
| Loss | Smooth L1 Loss |
| Loss $\beta$ | 5.0 |
| Rollouts per update | 24 |
| Learning epochs | 15 |
| Mini-batches | 4 |

Table 2: Skill discovery hyperparameters.

| Name | Objective | Weight |
|---|---|---|
| Collisions | $\mathbf{1}_{body/thigh/shank\ collision}$ | $30.\Delta_t$ |
| Feet air time | $(\mathbf{t}_{air} - 0.5)$ | $10.\Delta_t$ |
| Joint vel limits | $\begin{cases} -\|\dot{\mathbf{q}} - \dot{\mathbf{q}}_{\text{lim}}\| & \text{if } \dot{\mathbf{q}} \notin [-\dot{\mathbf{q}}_{lim}, \dot{\mathbf{q}}_{lim}] \\ 0 & \text{otherwise} \end{cases}$ | $-10.\Delta_t$ |
| Joint acceleration | $-\|\ddot{\mathbf{q}}\|^2$ | $-2.5\text{e-}7\Delta_t$ |
| Action rate | $-\|a_t - a_{t-1}\|^2$ | $-5.\text{e-}2\Delta_t$ |
| Energy | $-\|\dot{\mathbf{q}}\boldsymbol{\tau}\|$ | $-1.\text{e-}3\Delta_t$ |
| Nominal joint position | $-\|\mathbf{q} - \mathbf{q}_{nom}\|^2$ | $-2.\Delta_t$ |
| Orientation | $-\|\mathbf{g}\|^2$ | $-10.\Delta_t$ |
| Base height | $-\|p_z - 0.6\|^2$ | $-5.\Delta_t$ |

Table 3: Extrinsic reward terms and weights. All of the scales are multiplied by the time step size $\Delta_t$.

## A.5 ADDITIONAL COMPARISONS WITH BASELINE LSD AND METRA METHODS

In Fig. 9 we sample skills uniformly from a 2-D circle and linearly vary the magnitude, resulting in the pattern shown on the left (first column). We compare the latent space encodings (second column) and XY trajectories (third and fourth columns) between the baseline METRA (top), LSD (middle), and Ours (bottom). The baseline methods (LSD and METRA) only optimize for the alignment of latent transitions and skills and always maximize their magnitude. This results in the latent and XY state transitions being maximized for *all* skills, regardless of the magnitude of the original skill. On the real robot, this in turn translates to very high base linear velocities for every skill. On the other hand, our proposed method preserves the magnitude information in the latent space and enables the learning of behaviors with varying linear velocities, as can be seen from the bottom plots in Fig. 9. In Fig. 10 we show the evolution of the base position and base linear velocities for ours and one of the baseline methods for a zero-shot goal tracking task. As can be seen, the baseline overshoots the desired target several times and eventually falls over. In comparison, our approach slows down as it approaches the goal, stops and maintains its position. We further quantitatively evaluated the goal-tracking performance across 200 trials for LSD, METRA, and our approach (with Euclidean distance) in Fig. 11, in terms of the root mean squared error between the robot and its goal as a function of time. As can be seen, all methods approach the targets in a similar manner, but both METRA (orange) and LSD (green) overshoot to different extents. On the other hand, our method smoothly converges onto the target.

In Fig. 12 we compare the mean (across all parallel environments) and the maximum traveled distance in an episode throughout training. Due to the additional complexity of predicting the mag-

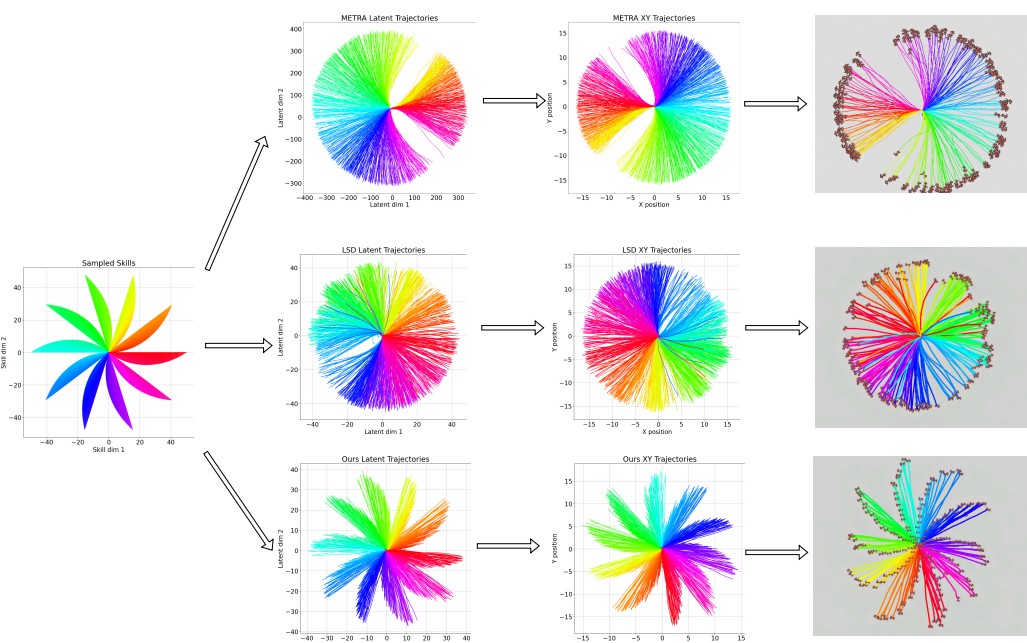

Figure 9: Comparison between the baseline METRA (top rows), LSD (middle rows), and Ours (bottom rows).

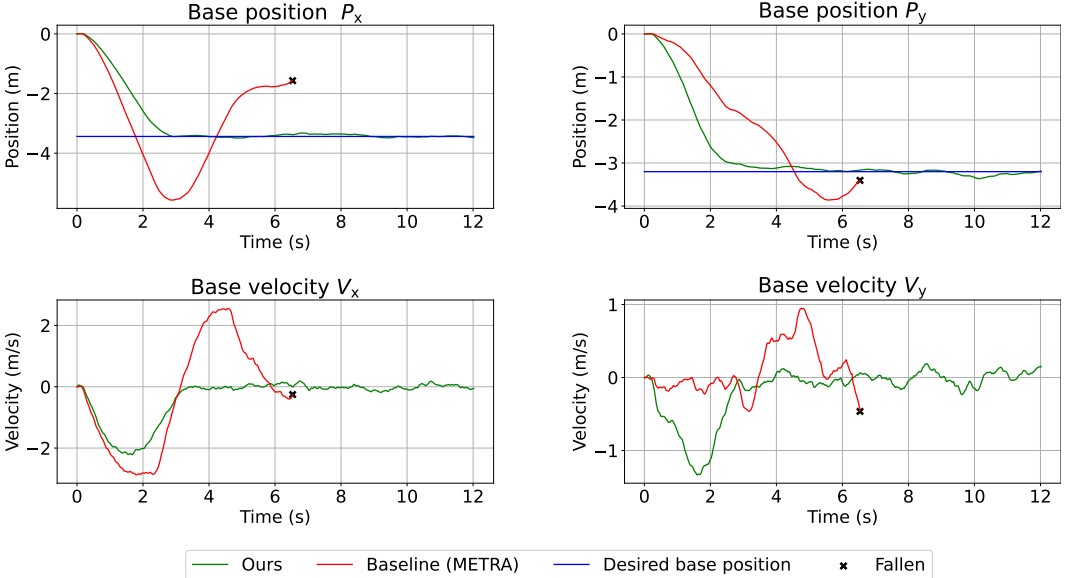

Figure 10: Comparison between base position (in the world frame) and base linear velocity (in the local frame) between the baseline METRA (in red) and Ours (in green) in simulation. The desired final position is shown by the blue line.

nitude, our method takes about 15k steps longer to converge. However, as can be seen from the left plot, our method has a lower mean traveled distance while maintaining the same maximum value as the baseline approaches, showing that it has learned a broader range of skills.

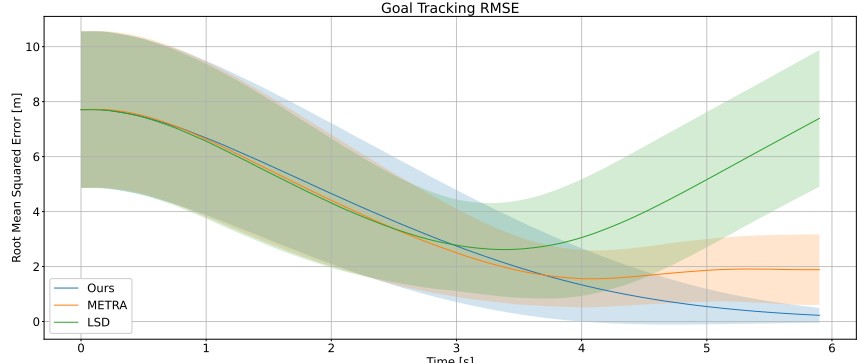

Figure 11: Quantitative evaluation of zero-shot goal-tracking performance across 200 trials for Ours (blue), METRA (orange) and LSD (green). The lines indicate the mean tracking error (across the trials) as a function of time, and the shaded regions show its $\pm$ std.

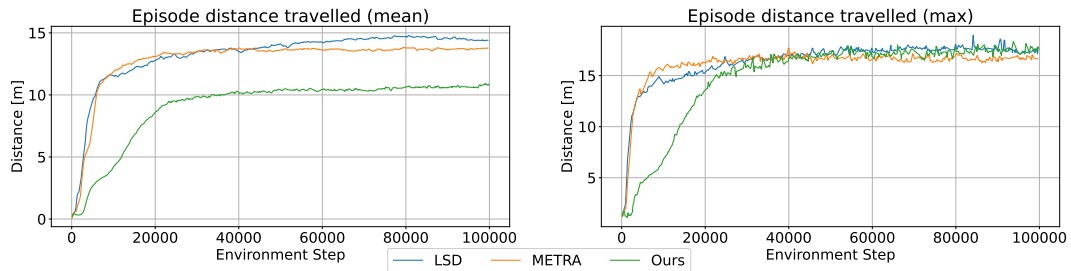

Figure 12: The mean (left) and max (right) traveled distance during an episode, as a function of the number of environmental steps throughout training. Ours (in green) can reach the same maximum distance, while keeping the distribution (and thus the range of skills) much broader.

### A.6 INTRINSIC AND EXTRINSIC REWARD ABLATIONS

The regularization rewards that we propose are necessary for a successful hardware deployment. However, they can add a degree of structure to the problem, which makes the task of skill discovery easier. To better evaluate the effect of our proposed algorithm (rather than that of the extrinsic regularization rewards), we compare several ablations:

- **Baseline** - Our proposed method.
- **Variant 1** - Intrinsic rewards only.
- **Variant 2** - Intrinsic rewards with body-contact termination.
- **Variant 3** - Intrinsic rewards with body-contact termination and contact avoidance extrinsic rewards.

As can be seen from Fig. 13, **Variant 1** which uses only the intrinsic rewards and no termination conditions has the lowest performing behavior. Qualitatively, the behavior consists of the robot falling down and rolling around a bit, resulting in the small maximum traveled distance. Just the addition of a termination condition on body-ground contacts (**Variant 2**) enables the robot to learn a good locomotive policy. For highly articulated robots like the ANYmal, after falling down it is very challenging to either move or recover to the nominal pose. Without any external bias, if the robot converges to this local optimum it becomes very hard for it to explore enough so that it learns to stand up and then locomote. Finally, by rewarding a lack of hip and shank contacts, as in **Variant 3**, we can get a much better policy with a similar loss to our proposed **Baseline**. The main differences between **Variant 3** and **Baseline** is that the former is much less robust and stable, and covers a smaller distance in each episode.

From this ablation it can be seen that while the extrinsic rewards can help give some helpful

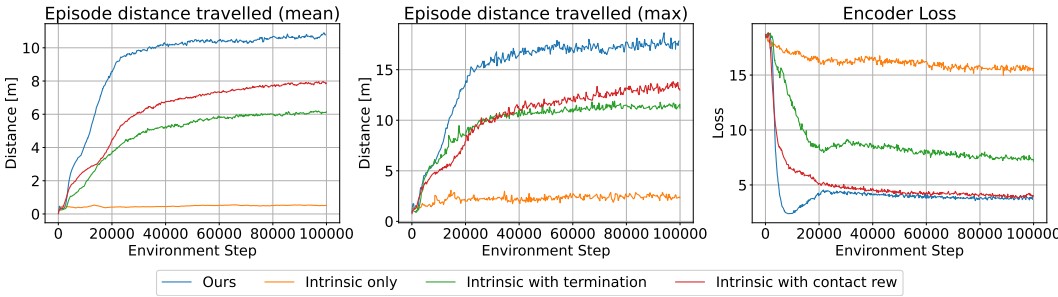

Figure 13: Ablation study between Ours (blue), intrinsic rewards only (orange), intrinsic rewards with termination on body contacts (green), and intrinsic rewards with termination on body contacts and extrinsic reward for avoiding hip/shank collisions (red).

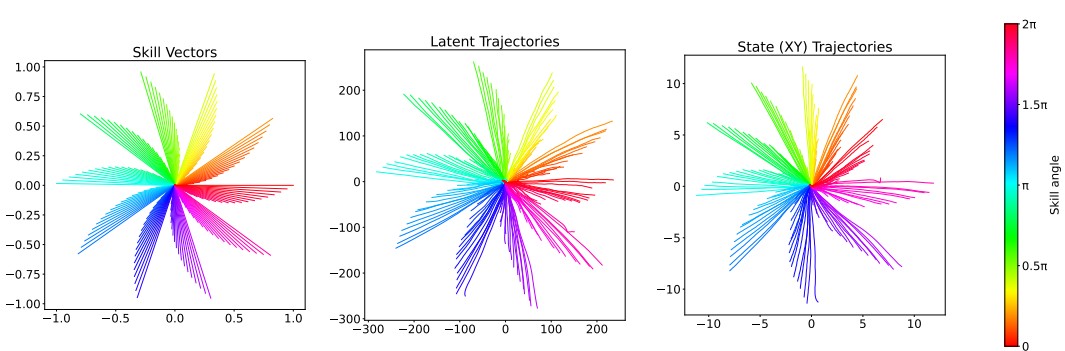

Figure 14: Locomotive behaviors learned with our method under a temporal distance constraint.

structure to the problem without constraining too strictly, they are not crucial for discovering locomotive skills (as shown by **Variant 2**). While **Variant 1** did not perform well, we believe that the main reason is due to the "prone" local minimum, from which the policy cannot easily escape. This agrees with prior work (Park et al., 2021) where an "upright base" bonus is used on the Humanoid environment to get good behaviors. METRA (Park et al., 2023b) does not use such a bonus, and the policy converges to a rolling behavior — which is not possible on the ANYmal-C due to its morphology, rather than locomotion.

### A.7 USING OTHER DISTANCE METRICS FOR THE CONSTRAINT

In this work, we used Euclidean distance as the constraint in the optimization objective, as shown in Eq. 10. While this works well for locomotive behaviors, one can expect that there are certain environments and tasks where Euclidean distance might be suboptimal. In literature, several works have proposed using other distance metrics, such as a controllability-aware distance Park et al. (2023a), temporal distance (METRA) Park et al. (2023b), or a language-based distance function Rho et al. (2024). The Temporal distance between two states $s_0$ and state $s_T$ is the minimum number of environment steps required to transition between them. To show that our approach can be combined with different distance metrics, we evaluate its performance using the temporal distance constraint as in METRA, and otherwise keep the same norm-matching objective. We constrain the magnitude of the skills to in the range $[0, 1]$[1] and set the constraint to be $||\phi(\mathbf{s}') - \phi(\mathbf{s})|| \leq 1$ as in METRA, which has been shown to be equivalent to $||\phi(\mathbf{s}_T) - \phi(\mathbf{s}_0)|| \leq d_{temporal}$. By selecting a magnitude of, for example, 0.5, we encourage the agent to achieve half the temporal distance achieved under a magnitude of 1. As can be seen in Fig. 14, our method can also achieve competitive performance

---

[1]During training we sample magnitudes in the range $[0, 1.5]$ rather than $[0, 1]$ as we found it biases the approach towards learning behaviors with larger temporal distances due to the implicit temporal distance maximization.

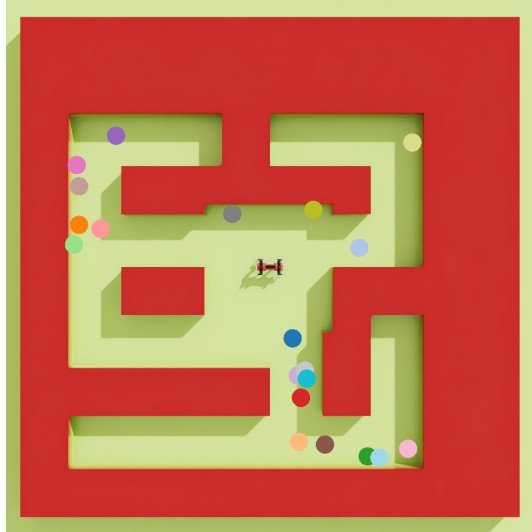

Figure 15: The maze environment with the walls shown in red. For the zero-shot goal-tracking experiment, we sample the shown points around the traversed parts of the maze.

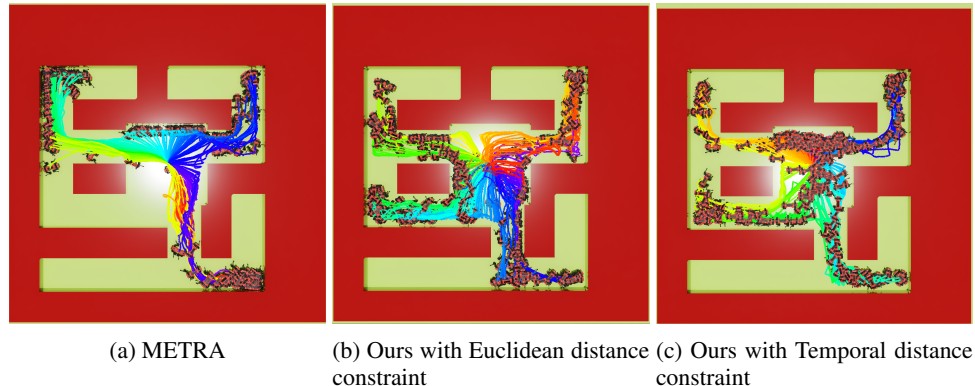

(a) METRA       (b) Ours with Euclidean distance constraint       (c) Ours with Temporal distance constraint

Figure 16: Learned skills in the maze environment with uniformly sampled skills (as in Fig. 1).

with temporal distance as the constraint. This result is important, as it shows that the norm-matching objective can learn a wider range of behaviors regardless of the chosen distance constraint.

### A.8 CAN WE LEARN STATE-TRAVERSING BEHAVIORS IN MORE COMPLEX ENVIRONMENTS?

In this section we evaluated the skills learned in complex maze environment (see Fig. 15), where the robot cannot simply locomote in arbitrary directions, using METRA, and our approach with both temporal and Euclidean distance for the constraint. We used the same training setup, with the agents initialized near the center of the maze with a longer episode duration of 20 seconds. We use the same observations for the encoder as in the rest of the experiments, but add the base position and height scans around the robot to the policy. As can be seen from Fig. 16, all three variations show relative success at traversing a large part of the maze. However, the behaviors learned by our method cover a more uniform area of the maze, unlike METRA where for all skills the agents saturate at different corners of the maze.

It is interesting to note that not all areas of the maze are visited by either method. A possible explanation is that the agent might have to traverse the same path under different skills to reach new areas of the maze. As this is suboptimal per-step (since it would make it difficult to align the latent encodings with the skills), the agent might never explore enough to discover the distant areas of the maze.

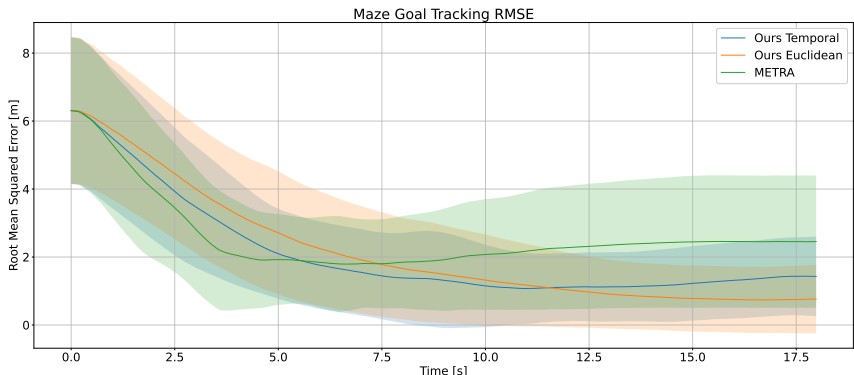

Figure 17: Zero-shot goal-tracking performance for METRA (green) and our method with a Euclidean (orange) and Temporal (blue) distance constraint. The lines show the mean value across the 20 trials, and the filled areas indicate the $\pm$ standard deviation.

We further test the zero-shot goal tracking in this new environment. In Fig. 15 we sample 20 desired goal points within the visited areas of the maze and use the same method as in Section 4.3 to select skills. Fig. 17 shows the RMSE as a function of time for the three methods (with the mean and standard deviation taken across the 20 goals). With both Euclidean or Temporal constraint, our method outperforms METRA, although the general tracking performance is worse than in the open space environment, likely due to the larger complexity of the environment.

