# OpenReview forum: "Constrained Skill Discovery: Quadruped Locomotion with Unsupervised Reinforcement Learning"
_ICLR.cc/2025/Conference — Submitted to ICLR 2025_

### Official Review · Reviewer_WVrN · 2024-10-16

**Soundness:** 3
**Presentation:** 4
**Contribution:** 3
**Rating:** 6
**Confidence:** 5

**Summary:**

The paper proposes a method for unsupervised reinforcement learning to enable quadruped robots to autonomously learn diverse locomotion skills. The key idea is to maximize the mutual information between the robot's skills and states while constraining the distance covered, replacing the previous latent transition maximization with a norm-matching objective. This allows for richer state space coverage and more controllable and stable locomotive behaviors than prior approaches. The method demonstrates successful deployment on the ANYmal quadruped robot, where the robot can accurately perform zero-shot goal tracking without any task-specific rewards, relying solely on intrinsic skill discovery.

**Strengths:**

- The paper is clearly presented, with abundant ablation studies and various baseline methods. The analysis in the experiment part explains well how the introduced constraint can help improve the state space coverage intuitively.
- The experimental setup is thorough, with comparisons against state-of-the-art methods like Lipschitz-constrained skill discovery (LSD) and Metric-aware abstraction (METRA). The paper presents extensive performance evaluations in both simulated and real environments using the ANYmal robot, showing robust results in real-world conditions.

**Weaknesses:**

- The proposed method additionally imposes a norm matching constraint on the original LSD formulation to regularize the agent from maximizing the latent space transition. In Eq.9, using L2 norm on the original state space makes sense in the simplified setting where only euclidian distance is considered and XY plan coverage is focused on. It may fail to generalize to more complex state space to diversity skills in to justify the choice of this bound. Think of diversifying in a joint space; it is not reasonable to impose such a bound with the difference between two consecutive joint positions since similar joint states can represent completely different robot gaits.
- Limited experiments on a simplified setting with euclidian space coverage. It would be super helpful and make the proposed method much stronger to show results in style/gait diversification on quadruped, or simulated settings such as maze navigation, where the state space euclidian distance does not reflect the adjacency in the latent space.

**Questions:**

- Regarding weakness 1: if LSD fails to regularize the norm of the latent space transition, why not simply add it back or normalize it to a unit vector maintaining only directional information? Alternatively, why not consider selecting state space such that it encodes \phi(s_t, s_{t+1}) or \phi(s_{t+1}-s_t) where you process the directional information before encoding?
- It would be helpful to demonstrate the applicability of the proposed method with experiments on more complex settings other than the single simplified euclidian space coverage presented in the paper.
- For a fair comparison with ASE and CASSI, one should select encoder input with the same features as in the proposed method. As the authors pointed out, if joint information is considered, the diversity on the XY plane coverage will deteriorate. And the encoder input should consider states of consecutive steps to match the usage in the proposed method, such as (pos_xy, pos_xy', ...).

---

> ### Author Response · Authors · 2024-11-21
>
> We would like to thank reviewer WVrN for their extensive and constructive feedback.
>
> **Weaknesses**:
>
> > 1. The proposed method additionally imposes a norm matching constraint on the original LSD formulation to regularize the agent from maximizing the latent space transition. In Eq.9, using L2 norm on the original state space makes sense in the simplified setting where only euclidian distance is considered and XY plan coverage is focused on. It may fail to generalize to more complex state space to diversity skills in to justify the choice of this bound. Think of diversifying in a joint space; it is not reasonable to impose such a bound with the difference between two consecutive joint positions since similar joint states can represent completely different robot gaits.
>
> 1. We would like to highlight that our original state space contains both joint information (position and velocity) and Cartesian base states. Indeed, Euclidean distance does not make sense in all tasks and scenarios, which is a limitation that LSD also describes. However, in principle, our norm-matching objective can work regardless of the distance metric chosen for the constraint. For example, METRA uses a temporal distance constraint which learns behaviors that maximize the minimum number of environmental steps required to reach $s_T$ from $s_0$. Such constraints can work better in environments where Euclidean distance is not a good indicator of good performance.
> In the updated manuscript we show our full objective under such a temporal distance constraint, and show that it successfully learns locomotive behaviors (Fig. 14 in the new version). Empirically, we observed that the temporal distance resulted in quicker convergence compared to the Euclidean one we use in the rest of the work. We hope that this shows that our objective can incorporate various distance metric constraints, some of which might be better suited than Euclidean distance for certain tasks and environments.
>
> > 2.  Limited experiments on a simplified setting with euclidian space coverage. It would be super helpful and make the proposed method much stronger to show results in style/gait diversification on quadruped, or simulated settings such as maze navigation, where the state space euclidian distance does not reflect the adjacency in the latent space.
>
> 2.  Thank you for these suggestions on strengthening our contributions! In the updated manuscript, we have added a variation of our norm-matching objective using a temporal distance constraint (in place of the Euclidean distance one), as the one used in METRA (see new Fig. 14). The constraint encourages the agent to learn behaviors which take more environmental steps to achieve, and the norm-matching ensures that for skills with small magnitude travel shorter temporal distances. We also train our policy in a 2D maze environment using both Euclidean distance and temporal distance, as seen in Fig. 15, 16, and 17. This enables us to also perform zero-shot goal-tracking where straight lines in skill space result in curved lines in real space, allowing the robot to navigate around the maze corners.
> We would like to mention that this family of skill discovery algorithms cannot learn different gaits in its present form - or rather, while different gaits might emerge at different velocities (which we observe in our approach to some extent) they cannot be the main distinguishing feature between skills. The reason for that is that LSD, METRA, and our approach optimize the skills and the latent transition between initial and final state (i.e. $||(\phi(s_T)-\phi(s_0))-z||^2$ (or the simplified version of $(\phi(s_T)-\phi(s_0))^Tz$ for LSD/METRA). This would not directly work for cyclical motions (like gaits) where the initial and final state might coincide. As future work we want to explore how such cyclical motions can be learned under this framework.

---

> ### Author Response · Authors · 2024-11-21
>
> **Questions**:
>
> > 1. Regarding weakness 1: if LSD fails to regularize the norm of the latent space transition, why not simply add it back or normalize it to a unit vector maintaining only directional information? Alternatively, why not consider selecting state space such that it encodes \phi(s_t, s_{t+1}) or \phi(s_{t+1}-s_t) where you process the directional information before encoding?
>
> 1. We thank the reviewer for the interesting discussion points. In essence, our method balances the discovery of meaningful state transitions (in our case locomotive skills) whilst limiting aggressive behaviors.
>
> Previous methods which are unconstrained such as ASE essentially optimize the objective proposed by the reviewer as a way of maximizing the KL divergence between skills and states. The problem is that the KL divergence is maximized when there’s no overlap between the two distributions - and there’s no advantage in pushing the distributions further. In practice, this results in ever-so-slightly distinctive behaviors (as shown in Fig. 4). The way LSD addresses this is to enforce a constraint on the latent transition magnitude to be upper bound by some distance metric (such as the Euclidean distance between real state transitions). The result is that the agent is encouraged to maximize the latent transition, and thus implicitly (through the constraint) maximize the chosen distance metric. If we choose to normalize the latent transition, we cannot enforce this constraint anymore (since the latent transition magnitude will always be 1).
>
> The issue essentially breaks down to the fact that this constrained maximization helps achieve meaningful state transitions, but always maximizing it (like LSD and METRA do) results in aggressive behaviors that we cannot deploy on real systems. In our approach, we maintain the constraint but force the encoder to match the norm of the skill rather than just its direction. The result is that large skills can result in large real state transitions, while we also maintain the ability to achieve behaviors with smaller real transitions by using skills with small magnitudes.
>
> >2.  It would be helpful to demonstrate the applicability of the proposed method with experiments on more complex settings other than the single simplified euclidian space coverage presented in the paper.
>
>
> Following the reviewer’s suggestion, we first show that our approach can successfully learn behaviors with non-Euclidean constraints as well - such as temporal distance (Section A.7 in the Appendix). We have also added a complex maze environment, which the robot has to learn to navigate (Section A.8).
>
> > 3. For a fair comparison with ASE and CASSI, one should select encoder input with the same features as in the proposed method.
>
> As the authors pointed out, if joint information is considered, the diversity on the XY plane coverage will deteriorate.
> We would like to highlight that for ASE, CASSI and DOMiNiC we compare to both a version using the same features as our proposed method (shown in Fig. 4 as “ASE 2D”, “CASSI 6 skills”, and “DOMiNiC 16 skills”, respectively), and to a reduced set of features (labeled with “XYO”), which ONLY contain the base position and/or base velocities. Our method (together with LSD and METRA) uses the full set of observations, including the joint positions and velocities. In the updated manuscript, we explained these annotations in Fig. 4.
>
> > And the encoder input should consider states of consecutive steps to match the usage in the proposed method, such as (pos_xy, pos_xy', ...).
>
> Thank you for the suggestion. We chose to use the original formulation for the encoder in ASE (i.e. $\phi(s,s’)^Tz$) , but as the reviewer pointed out this might bias the results, so we will include a comparison using ASE with $(\phi(s’)-\phi(s))^Tz$ as the objective in the updated version.

---

> ### Author Response · Authors · 2024-11-26
>
> Dear Reviewer WVrN,
>
> We sincerely appreciate your insightful comments and valuable suggestions on improving our work. As the final revision deadline approaches, we would kindly like to ask whether our responses have sufficiently addressed your questions and concerns. We are happy to engage in further discussion or provide any clarifications needed, and we welcome any additional feedback to strengthen our work before the rebuttal period ends.
>
> Thank you for your time and thoughtful reviews.
>
> Kind regards,
>
> Authors

---

> > ### Comment · Reviewer_WVrN · 2024-11-27
> > **Thank you for addressing the concerns**
> >
> > Thank you for addressing the concerns and questions raised. I have raised my score to 6. I am not convinced I should raise the score to a higher one since the experiments conducted in the paper are still limited to 2D space coverage scenarios. The paper's conclusion can be further reinforced with results on more complex state spaces.

---

### Official Review · Reviewer_yEir · 2024-10-31

**Soundness:** 3
**Presentation:** 3
**Contribution:** 2
**Rating:** 5
**Confidence:** 3

**Summary:**

This paper proposes a novel unsupervised reinforcement learning method for skill discovery, which improves upon previous constrained skill discovery methods (LSD[1], METRA[2]) by introducing a norm-matching objective. This norm-matching objective replaces latent transition maximization, achieving the following goals: (1) enabling the agent to learn a broader range of skills and reliably cover a larger portion of the state space; (2) addressing the issue of overly aggressive, high-speed movements resulting from the previous maximization objective, which are challenging to deploy on most robotic hardware. Experiments demonstrate that, compared to baseline methods, this approach allows for more accurate target-tracking in quadruped robots and enables zero-shot deployment on actual quadruped hardware.

**Strengths:**

1. The paper is clearly written, with Sections 2 and 3 providing a well-articulated motivation and a strong connection to prior work. Both the method and empirical studies are explained with considerable detail.

2. The proposed approach allows for skill control in terms of both magnitude and direction, enabling the robot to navigate the environment at different speeds.

3. The experimental results are impressive, particularly the position-tracking comparisons. Compared to previous methods, this approach appears to achieve more accurate position and velocity tracking without relying on task-specific domain knowledge.

**Weaknesses:**

- Previous methods [1][2] conducted comparative experiments on multiple benchmark tasks (e.g., Ant, HalfCheetah) and validated their effectiveness across various downstream tasks. This paper, however, only conducts experiments on the ANYmal quadruped robot, focusing primarily on velocity tracking. (If this paper aims to demonstrate the broad applicability of the proposed method, comparison experiments should be conducted on datasets like Ant and HalfCheetah. If the focus is on quadruped robots, it is necessary to compare the method with other controllers designed for quadruped robots.)

**Questions:**

- The derivation in Equations 6-7 of the paper contains errors; the coefficients after expanding the squared difference formula are incorrect. What is the purpose of comparing the magnitudes of \( L \) and \( L^{LSD} \)? Compared to \( L \), \( L^{LSD} \) is a simplified result. Does choosing the unsimplified \( L \) as the optimization function make the optimization more complex?
- Figure 7 shows that the ANYmal robot can accurately track velocity commands in the x and y directions. What specific performance criteria were used to evaluate tracking in this experiment? (e.g., The experiment can provide quantitative metrics, such as the mean squared error of position or velocity over time, and compare the performance with controllers that include velocity tracking rewards or controllers from previous methods.)
- In the final 30 seconds of the supplementary video, the test results on the ANYmal robot are demonstrated. However, this section lacks textual and verbal explanations, making it unclear what the paper intends to convey. (e.g., The unannotated parts of the video could include annotations displaying the target position or velocity, or show comparative content.)

[1] Seohong Park, Jongwook Choi, Jaekyeom Kim, Honglak Lee, and Gunhee Kim. Lipschitzconstrained Unsupervised Skill Discovery. October 2021.
[2] Seohong Park, Oleh Rybkin, and Sergey Levine. METRA: Scalable Unsupervised RL with Metric-Aware Abstraction. October 2023.

---

> ### Author Response · Authors · 2024-11-21
>
> We would like to thank reviewer yEir for their feedback and suggestions on improving our work.
>
> **Weaknesses**:
>
> > 1. Previous methods [1][2] conducted comparative experiments on multiple benchmark tasks (e.g., Ant, HalfCheetah) and validated their effectiveness across various downstream tasks. This paper, however, only conducts experiments on the ANYmal quadruped robot, focusing primarily on velocity tracking. (If this paper aims to demonstrate the broad applicability of the proposed method, comparison experiments should be conducted on datasets like Ant and HalfCheetah. If the focus is on quadruped robots, it is necessary to compare the method with other controllers designed for quadruped robots.)
>
> Thank you for your feedback! As mentioned in our contributions, our focus was specifically on using skill discovery for learning quadruped robot skills. This motivated our choice of experiments, such as goal-tracking.
> We would like to emphasize that both LSD [1] and METRA [2] only show single and multiple goal tracking (e.g. AntGoal and AntMultiGoals) as downstream locomotive tasks in the continuous skill case; and HalfCheetahHurdle, AntRotation and AntQuaternion as non-locomotive downstream tasks for their discrete locomotive skills.
> Similarly, in this work we compare the zero-shot goal (position rather than velocity) tracking with LSD and METRA (in Figure 6, 10, and 11), which is analogous to the AntGoal environment but for a more complex ANYmal robot with more degrees of freedom. In Figure 8 we also show position and yaw tracking (which is a combination of AntGoal and AntRotation). Furthermore, we do this in a zero-shot way unlike LSD which trains it as a downstream task. We have also added a new maze environment (Appendix A.8), where the robot has to learn to walk around the walls of the 2D maze. We show zero-shot goal-tracking in this environment as well, and report lower RMSE compared to METRA.
>
> We wanted to avoid directly comparing with a reward-dense method, which are commonly used in quadruped locomotion literature. We can expect that reward-tuning would yield a “better” (for the task of goal-tracking) controller, but requires domain knowledge and tedious reward tuning. Instead, we show that unsupervised RL can be used to achieve such tasks with good performance, while requiring little domain knowledge or domain-specific task rewards.
>
> **Questions**:
>
> > 1. The derivation in Equations 6-7 of the paper contains errors; the coefficients after expanding the squared difference formula are incorrect.
>
> 1. We thank the reviewer for spotting this typo. The correct equation should read:
>
> $$-\underbrace{\frac{1}{2}\mathbb{E}\_{s,z}[(\phi(s_T)-\phi(s_0))^T(\phi(s_T)-\phi(s_0))]}\_\text{L2 Regularization} + \underbrace{\mathbb{E}\_{s,z}[(\phi(s_T)-\phi(s_0))^Tz]}\_\text{Directional Alignment} - \underbrace{\frac{1}{2}\mathbb{E}\_{s,z}[z^Tz] + c}\_\text{Constants}$$
>
> We would like to emphasize that this is a typological error only. We will correct it in the manuscript.
>
> > What is the purpose of comparing the magnitudes of ( L ) and ( L^{LSD} )? Compared to ( L ), ( L^{LSD} ) is a simplified result. Does choosing the unsimplified ( L ) as the optimization function make the optimization more complex?
>
> This is a great point - indeed this simplification can help make optimization easier, and this can be seen in Figure 11 in our original manuscript (Supplementary, now Figure 12 in the updated version) where LSD/METRA reach the peak performance earlier. Intuitively, the simplified objective only has to align the direction of the latent transition to that of the skill (and then maximize the magnitude of the transition). On the other hand, the full objective has to both align with the skills and match the magnitude.

---

> ### Author Response · Authors · 2024-11-22
>
> > 2. Figure 7 shows that the ANYmal robot can accurately track velocity commands in the x and y directions. What specific performance criteria were used to evaluate tracking in this experiment? (e.g., The experiment can provide quantitative metrics, such as the mean squared error of position or velocity over time, and compare the performance with controllers that include velocity tracking rewards or controllers from previous methods.)
>
> 2. We thank the reviewer for the suggestions. We would first like to clarify that Figure 7 shows position tracking rather than velocity tracking. As LSD/METRA were not safe enough to deploy on the hardware (as shown in Fig. 6 or in the video), we only compare the performance in simulation. In Fig. 10 in the Supplementary of the original manuscript we also have a simulation-based comparison between our approach and LSD. Following the reviewer’s suggestion, we performed a quantitative evaluation in terms of the position-tracking mean squared error as a function of time across 200 trials in sim, as shown in the new Fig. 11. We believe this better illustrates the differences in performance. We would like to mention that this analysis does not include failures (i.e. falling down), which were much more common with the baselines compared to our approach, due to their more aggressive behaviors.
>
> > 3. In the final 30 seconds of the supplementary video, the test results on the ANYmal robot are demonstrated. However, this section lacks textual and verbal explanations, making it unclear what the paper intends to convey. (e.g., The unannotated parts of the video could include annotations displaying the target position or velocity, or show comparative content.)
>
> 3. We appreciate this feedback about the clarity of the final section in the video. Our goal with this section was to show several more goal-tracking experiments (similarly to the one shown at the very start of the video). In the updated version we will improve clarity by showing the position targets.

---

> ### Author Response · Authors · 2024-11-26
>
> Dear Reviewer yEir,
>
> We sincerely appreciate your insightful comments and valuable suggestions on improving our work. As the final revision deadline approaches, we would kindly like to ask whether our responses have sufficiently addressed your questions and concerns. We are happy to engage in further discussion or provide any clarifications needed, and we welcome any additional feedback to strengthen our work before the rebuttal period ends.
>
> Thank you for your time and thoughtful reviews.
>
> Kind regards,
>
> Authors

---

> > ### Comment · Reviewer_yEir · 2024-11-29
> > **Response**
> >
> > Thank you for your explanation. I still have the following question:
> >
> > > We wanted to avoid directly comparing with a reward-dense method, which are commonly used in quadruped locomotion literature. We can expect that reward-tuning would yield a “better” (for the task of goal-tracking) controller, but requires domain knowledge and tedious reward tuning.
> >
> > While unsupervised RL has the advantage of avoiding cumbersome reward tuning, can it compete with reward-intensive methods, especially in complex tasks？

---

> ### Author Response · Authors · 2024-12-02
>
> >Thank you for your explanation. I still have the following question:
> We wanted to avoid directly comparing with a reward-dense method, which are commonly used in quadruped locomotion literature. We can expect that reward-tuning would yield a “better” (for the task of goal-tracking) controller, but requires domain knowledge and tedious reward tuning.
> While unsupervised RL has the advantage of avoiding cumbersome reward tuning, can it compete with reward-intensive methods, especially in complex tasks？
>
>
> Thank you for the question! While we already show significantly better tracking performance compared to baseline skill discovery methods, we have also included two baseline reward-dense methods. To summarise, our method achieves **lower tracking error** than even the reward-intensive methods, which we attribute to the learning of the diverse set of both low- and high-velocity behaviors. We would emphasize that our method does this zero-shot in an unsupervised way and does not require task-rewards unlike the reward-dense baselines.
>
>
> The first of the new baselines trains a goal-conditioned locomotive policy from scratch with two rewards based on the distance to the goal and a reward for velocity towards the goal. The second trains a high-level goal-conditioned policy on top of a pretrained low-level velocity policy with the same dense rewards as above. We show the tracking results below for 200 targets in the range [-10, 10] m for both x and y, in a 6-second episode. We report the data as the mean error and standard deviation (over the trials), averaged over the last 50 steps of the episode:
>
> | Method / Distance range(m) | 0.00m-4.52m | 4.52m-9.04m | 9.04m-13.56m |
> |:---:|:---:|:---:|:---:|
> | METRA | 1.49m ± 0.70 | 2.20m ± 1.33 | 1.65m ± 1.00 |
> | LSD | 9.66m ± 1.96 | 6.68m ± 1.37 | 3.98m ± 1.42 |
> | Dense reward | 0.12m ± 0.30 | 0.62m ± 0.92 | 4.28m ± 3.27 |
> | Pretrained dense reward | 0.23m ± 0.14 | 0.62m ± 0.58 | 2.48m ± 1.49 |
> | *Ours* | **0.04m ± 0.02** | **0.18m ± 0.12** | **0.79m ± 0.52** |
>
> As can be seen from the table, in this case our method outperforms both the skill discovery (SD) baselines, and the reward dense methods, especially for more distant goals, and with lower variance. We attribute this to the fact that our policy has learned both low- and very high-velocity skills, the former in contrast with SD baselines, and the latter with the dense reward baselines (which is why they track close goals well, but fail to reach the distant ones).
>
> We believe that this serves as sufficient evidence that our method is at least competitive with reward-dense controllers. While we did spend some time tuning the weights of these baselines, we would assume that with further reward tuning (and designing extra terms), the baselines might reach the same (or better) performance. However, that would come at the cost of significant effort and time.
>
> Thank you again for your feedback and please do let us know if you have any other questions.

---

### Official Review · Reviewer_A685 · 2024-11-04

**Soundness:** 4
**Presentation:** 2
**Contribution:** 2
**Rating:** 5
**Confidence:** 3

**Summary:**

This paper presents a method for constrained skill discovery applied to quadrupedal robot locomotion using unsupervised reinforcement learning. The proposed approach builds upon previous works that use mutual information between skills and states. In contrast to existing works, this paper introduces a norm-matching objective so as to achieve broader state space coverage with stable, controllable movements. This enables a quadruped robot to reach arbitrary points in its environment without requiring task-specific rewards. The proposed method was evaluated on the ANYmal robot.

**Strengths:**

- Based on the results, the norm-matching objective allows the robot to cover a broader range of states compared to previous approaches.
- It eliminates the need for task-specific rewards by using intrinsic motivation through skill discovery.
- The paper includes successful experiments with an actual robot.

**Weaknesses:**

- The novelty of the paper is questionable. It is obvious from the results that the norm-matching objective is indeed helpful, but this is just a simple regularization applied on top of existing methods. It is also the straightforward / obvious solution to the problem the paper talks about. Thus, it may not attract much interest.
- The paper is sometimes very difficult to follow. To me, this is a major concern that is sufficient for my rejection rating, because I believe the paper needs a major rewrite which is beyond a rebuttal. Starting from the second section, it uses a lot of notation and assumes some prior knowledge about skill discovery methods. This makes the paper inaccessible for people who are not working on this topic. A proper notation definition is missing, e.g., the variable z is used for skills starting from Section II but they are not given a formal definition as a mathematical object in the paper.

**Questions:**

I understand the problem the norm-matching objective is trying to solve. Did existing works try other approaches to solve the same problem and fail? Why is norm-matching interesting / novel? To me, it seemed like the most obvious solution.

---

> ### Author Response · Authors · 2024-11-22
>
> We would like to thank reviewer A685 for their feedback and for helping us improve the clarity of our work.
>
> **Weaknesses**:
>
> > 1. The novelty of the paper is questionable. It is obvious from the results that the norm-matching objective is indeed helpful, but this is just a simple regularization applied on top of existing methods. It is also the straightforward / obvious solution to the problem the paper talks about. Thus, it may not attract much interest.
>
> 1. Thank you for the feedback. We would like to clarify that our contributions include proposing this updated objective for skill discovery, and applying it to the task of quadruped locomotion on a real and complex robotic system. This allowed us to do safe and realistic zero-shot goal tracking on quadruped robots, which in literature typically requires reward-dense methods. We hope that our promising results will encourage more use of intrinsic motivation and unsupervised RL in the robotics domain, in particular.
> While mathematically the norm-matching objective breaks down to regularization, we emphasize that it results in significantly different and much more useful behaviors compared to the simplified objective.
>
> > 2. The paper is sometimes very difficult to follow. To me, this is a major concern that is sufficient for my rejection rating, because I believe the paper needs a major rewrite which is beyond a rebuttal. Starting from the second section, it uses a lot of notation and assumes some prior knowledge about skill discovery methods. This makes the paper inaccessible for people who are not working on this topic. A proper notation definition is missing, e.g., the variable z is used for skills starting from Section II but they are not given a formal definition as a mathematical object in the paper.
>
> 2. Thank you for the feedback regarding the clarity of the manuscript. We have rewritten parts of the manuscript to include more accessible explanations, including a description of the MDP and problem notations in our methodology (see section 2.1). We believe our paper is much more accessible in its updated version. Please let us know if there are any other specific sections which are hard to follow so that we can improve readability.
>
> **Questions**:
>
> > 1.  I understand the problem the norm-matching objective is trying to solve. Did existing works try other approaches to solve the same problem and fail? Why is norm-matching interesting / novel? To me, it seemed like the most obvious solution.
>
> 1. In general, few works apply these methods to real robotic systems, which are inherently more complex than some simulated benchmarks and it is where these issues are most prominent. Our proposed approach can discover agile, realistic and safe locomotion behaviors to be deployed on the hardware with purely skill discovery. Furthermore, the maximization problem occurs when using objectives like the one in LSD and METRA, due to the use of a Lipschitz constraint. Without the constraint, however, the agents learn much simpler behaviors, most of which are not useful for robotic control (as shown by both the original LSD/METRA works, and in Figure 4 in our manuscript).
> To the best of our knowledge, we are not aware of other works that try to address this issue. Some works (like CASSI and DOMiNiC) do apply other skill discovery methods to quadruped robots, but they combine them with either imitation learning or task-specific rewards which constrain the skill discovery to search around the optimal task policy.

---

> ### Author Response · Authors · 2024-11-26
>
> Dear Reviewer A685,
>
> We sincerely appreciate your insightful comments and valuable suggestions on improving our work. As the final revision deadline approaches, we would kindly like to ask whether our responses have sufficiently addressed your questions and concerns. We are happy to engage in further discussion or provide any clarifications needed, and we welcome any additional feedback to strengthen our work before the rebuttal period ends.
>
> Thank you for your time and thoughtful reviews.
>
> Kind regards,
>
> Authors

---

### Official Review · Reviewer_z7iM · 2024-11-08

**Soundness:** 2
**Presentation:** 2
**Contribution:** 2
**Rating:** 5
**Confidence:** 3

**Summary:**

This paper extends the Lipschitz-constrained Skill Discovery (LSD) framework by Park et al. (2022) to enable the learning of stable and controllable locomotion skills with broader state-space coverage. The proposed approach learns a skill-conditioned policy, where each skill is represented by a 2D continuous vector. This method allows robots to navigate at varying velocities without relying on task-specific rewards, instead using intrinsic mutual information and extrinsic regularization rewards. The authors further demonstrate that the policy can achieve effective zero-shot goal-tracking performance in quadruped locomotion tasks involving the ANYmal robot.

**Strengths:**

- The figures are well-illustrated and nicely done. I especially like Figures 1 & 4, which show a representation of skills in the 2D space.
- Real robot experiments using an ANYmal quadruped

**Weaknesses:**

1. The contributions are unclear. Could the authors clarify how this work improves upon LSD (Park et al., 2022)? Specifically, could you present a side-by-side comparison of the two objectives, highlighting what has been added or removed, and explain the rationale behind these changes?
2. It appears that this approach may not perform well in environments with obstacles. Based on the illustration of the sampled skills, it seems that the quadruped is capable of reaching any 2D location on a plane, but only in the absence of obstacles.
3. The authors overlook many relevant works related to skill discovery, including references [1-7]. Note that these works may use different terminology, such as 'options,' to achieve similar objectives.
4. The term 'skill' is not clearly defined. What is the duration of a skill? Does a skill have a termination condition? What constitutes a skill?
5. While I appreciate the demonstration of results on a real robot, it would also be valuable to see results on simulated benchmarks to better illustrate the efficacy of the proposed approach compared to LSD and Metra.

**References**
1. Durugkar, I., Tec, M., Niekum, S., & Stone, P. (2021). Adversarial intrinsic motivation for reinforcement learning. Advances in Neural Information Processing Systems, 34, 8622-8636.
2. Cho, D., Lee, S., & Kim, H. J. (2023). Outcome-directed reinforcement learning by uncertainty & temporal distance-aware curriculum goal generation. arXiv preprint arXiv:2301.11741.
3. Li, C., Song, D., and Tao, D. Hit-mdp: learning the smdp option framework on mdps with hidden temporal embeddings. In The Eleventh International Conference on Learning Representations, 2022.
4. Richard S Sutton, Doina Precup, and Satinder Singh. Between mdps and semi-mdps: A framework for temporal abstraction in reinforcement learning. Artificial Intelligence, 112(1-2):181–211, 1999
5. Pierre-Luc Bacon, Jean Harb, and Doina Precup. The option-critic architecture. In Thirty-First AAAI Conference on Artificial Intelligence, 2017
6. Jean Harb, Pierre-Luc Bacon, Martin Klissarov, and Doina Precup. When waiting is not an option: Learning options with a deliberation cost. In Thirty-Second AAAI Conference on Artificial Intelligence, 2018
7. Nicholas K Jong, Todd Hester, and Peter Stone. The utility of temporal abstraction in reinforcement learning. In AAMAS (1), pp. 299–306. Citeseer, 2008.

**Questions:**

1. In Equation 10, it seems there may be a misuse of notation. Is \( N \) a scalar rather than a function? If so, is this simply a multiplication?
2. How does this approach handle practical environments with obstacles?
3. What is the rationale behind restricting the latent space to a 2D vector? Could you provide some intuition on when this restriction might be beneficial and when it might need to be extended?
4. In Figure 3, the density distribution of the proposed method appears similar to a uniform random distribution. Why is this desirable? Does it indicate greater coverage of the state space?

---

> ### Author Response · Authors · 2024-11-22
>
> We would like to thank reviewer z7iM for their feedback and interesting discussion points.
>
> **Weaknesses**:
>
> > 1. The contributions are unclear. Could the authors clarify how this work improves upon LSD (Park et al., 2022)? Specifically, could you present a side-by-side comparison of the two objectives, highlighting what has been added or removed, and explain the rationale behind these changes?
>
> 1. We appreciate the reviewer’s feedback on the clarity of our contributions. We will make our contributions clearer in the updated manuscript.  In essence, prior works like LSD align and always maximize the latent transition. This can be an issue for many robotic applications, as it results in faster and more aggressive motions. In contrast, we want to limit aggressive motions to be able to safely deploy the controller on real robots.
>
> On the other hand, the full objective attempts to align and match the magnitude of the skills and the latent transition. Since the latent transition is upper bound by the chosen distance metric, we can essentially ensure that skills with low magnitude will result in small latent transitions, and crucially, would not push the real transitions to be large (due to the Lipschitz constraint).
>
> This can be seen in the equation below:
> $$L = -\frac{1}{2}\mathbb{E}\_{s,z}[||(\phi(s\_T)-\phi(s\_0))-z||^2] + c $$
> $$= -\underbrace{\frac{1}{2}\mathbb{E}\_{s,z}[(\phi(s\_T)-\phi(s\_0))^T(\phi(s\_T)-\phi(s\_0))]}\_\text{L2 Regularization} + \underbrace{\mathbb{E}\_{s,z}[(\phi(s\_T)-\phi(s\_0))^Tz]}\_\text{Directional Alignment} - \underbrace{\frac{1}{2}\mathbb{E}\_{s,z}[z^Tz] + c}\_\text{Constants}$$
> $$ L^{LSD} =  \underbrace{\mathbb{E}\_{s,z}[(\phi(s\_T)-\phi(s\_0))^Tz]}\_\text{Directional Alignment} $$
>
> LSD and METRA only optimize the directional alignment component, while they also enforce a 1-Lipschitz constraint on the latent transition $||\phi(s')-\phi(s)||$.
>
> In our work, we instead maximize the full objective ( second Equation above, i.e. $L = -\frac{1}{2}\mathbb{E}\_{s,z}[||(\phi(s\_T)-\phi(s\_0))-z||^2]$), while also keeping the constraint on the latent transition.
>
>
>
> > 2. It appears that this approach may not perform well in environments with obstacles. Based on the illustration of the sampled skills, it seems that the quadruped is capable of reaching any 2D location on a plane, but only in the absence of obstacles.
>
> 2. This is an interesting point to discuss. It is true that we train the skill learning framework in the absence of obstacles - and thus the learned behaviors themselves correspond to locomotion in uniform directions.
>
> In our view, the strength of unsupervised skill discovery is most prominent when learning general task-agnostic behaviors that can be reused for various downstream tasks. From this perspective, navigating around obstacles is seen as a planning task - for example, one could easily train a high-level skill-selecting policy to use the existing skills and avoid obstacles in the environment. On the other hand, if we learn low-level skills specifically to navigate a given maze, for example, these behaviors would then not be as general and useful for downstream tasks in other types of environments - be it either different mazes or even an open 2D plane.
>
> With that in mind, we were also interested in seeing whether the robot can learn interesting behaviors in a more complex maze environment. We tested this in a relatively complex 2D maze (environmental details are in Appendix Section A.8). In the updated version, we show that our approach does learn to traverse a large portion of the maze (Fig. 16 b) and c)). Furthermore, in Fig. 17 we show that our method can also perform zero-shot goal-tracking around the maze walls relatively well by using the learned skills.
>
> > 3. The authors overlook many relevant works related to skill discovery, including references [1-7]. Note that these works may use different terminology, such as 'options,' to achieve similar objectives.
>
> 3. Thank you for bringing these relevant works to our attention. Indeed the literature uses different terms for very similar ideas, such as options, empowerment and skill discovery. We have reviewed the relevant literature and will add the appropriate references in the updated manuscript.

---

> ### Author Response · Authors · 2024-11-22
>
> > 4. The term 'skill' is not clearly defined. What is the duration of a skill? Does a skill have a termination condition? What constitutes a skill?
>
> 4. In this case, a skill (denoted by z) is a random variable that encodes different behaviors. In practice, we use continuous N-dimensional vectors (where we show N=2 and N=3) and condition the policy on the state and the skill $\pi(a|s,z)$. During training, we choose to use one skill per episode, where we sample the skill from a prior distribution (uniform) at the start of the episode, and keep it fixed until the episode ends (either due to a time-out or a termination due to body contact). We have included some of these descriptions in the updated manuscript, together with a further definition of the MDP and variable names we use throughout the paper.
>
> > 5. While I appreciate the demonstration of results on a real robot, it would also be valuable to see results on simulated benchmarks to better illustrate the efficacy of the proposed approach compared to LSD and Metra.
>
>
> 5. We thank the reviewer for their feedback. We would like to highlight that we directly compare our method to LSD and METRA in simulation in Figures 4, 6, 9 and 10 on the ANYmal robot. In our opinion, ANYmal is a much more complex real-life system than most simulated benchmarks, with a higher number of degrees of freedom, underactuated coupled dynamics and significant mass and inertia. In this work, our goal was to show that unsupervised skill discovery can be used to achieve useful tasks on real robots and we hope that our promising results will encourage more use of these methods within robotics.
>
>
> **Questions**:
>
> > 1. In Equation 10, it seems there may be a misuse of notation. Is ( N ) a scalar rather than a function? If so, is this simply a multiplication?
>
> Thanks for spotting this, indeed this is meant to be a multiplication by a scalar N (which is the number of steps per episode). We will fix this in the manuscript.
>
> > 2. How does this approach handle practical environments with obstacles?
>
> As mentioned above, we have trained our policy in a 2D maze environment. In the updated manuscript, we demonstrate the effectiveness of our proposed approach in such an environment as shown in the (new) Figures 14,15 and 16.
>
> > 3. What is the rationale behind restricting the latent space to a 2D vector? Could you provide some intuition on when this restriction might be beneficial and when it might need to be extended?
>
> We first chose 2D latent space as that was used in LSD and METRA, which we built upon. In addition, as we are trying to learn locomotive skills for a quadruped robot, 2D skill space also makes sense for achieving position control. In LSD, it was reported that increasing the dimensions further (up to 5D on Ant) would still collapse to learning 2D locomotive behaviors. In our work, we notice that increasing to 3D (as shown in Fig. 8) results in additional yaw-orientation control.
>
> Since quadruped robots are underactuated with coupled dynamics, it is in general not possible to control each degree of freedom independently - for example the body’s motion depends on the motion of each leg and the interaction with the ground. If we would like to extend this approach to loco-manipulation using a quadruped robot with a manipulator arm, a higher skill dimension would be necessary to control both the base and the arm.
>
> > 4. In Figure 3, the density distribution of the proposed method appears similar to a uniform random distribution. Why is this desirable? Does it indicate greater coverage of the state space?
>
> To clarify, Figure 3 shows the density of mean base velocities (with the mean taken across an episode) using uniformly sampled skills. With LSD and METRA, each skill essentially consists of accelerating from 0 to the (nearly) max velocity - as shown by the histogram.
>
> Measuring the state coverage is complicated, as both of these methods still transiently visit the low-velocity areas (as they accelerate towards the high velocity ones). In contrast, with our method the agent visits and stays in these low-velocity areas for certain skills. This is desirable because it gives you much more fine-grained control on downstream tasks, such as the position goal-tracking we show in Figure 6.
>
> With LSD you essentially have a bang-bang controller, which causes it to overshoot the target repeatedly (see Fig. 6 and 10 for example) - since it has no skills that result in slow motion that can make it reach the target and stop.

---

> > ### Author Response · Authors · 2024-11-26
> >
> > Dear Reviewer z7iM,
> >
> > We sincerely appreciate your insightful comments and valuable suggestions on improving our work. As the final revision deadline approaches, we would kindly like to ask whether our responses have sufficiently addressed your questions and concerns. We are happy to engage in further discussion or provide any clarifications needed, and we welcome any additional feedback to strengthen our work before the rebuttal period ends.
> >
> > Thank you for your time and thoughtful reviews.
> >
> > Kind regards,
> >
> > Authors

---

### Author Response · Authors · 2024-11-25
**List of changes**

We would like to sincerely thank all of the Reviewers for their extensive feedback and suggestions on how to improve our work. We believe the changes based on the Reviewers' feedback have significantly improved the quality of our manuscript and strengthened our contributions.

You can find the **full updated rebuttal manuscript under PDF**, and **a version highlighting the changes in blue under Supplementary Materials -> diff_submission.pdf**, together with an **updated video**.

Below we will highlight the main changes done throughout the rebuttal process:

1. We additionally train our proposed method with a Temporal distance metric constraint (as in METRA) and show successful learning of locomotive behaviors. This shows that our approach is not limited to Euclidean constraints and can scale to different distance metrics (Sec. A.7 with Figure 14).

2. We have added a new 2D maze environment and show a comparison of the learned skills between our method (using both Euclidean and Temporal distances), and METRA (Sec. A.8 with Figures 15-17). We further show zero-shot goal-tracking in this maze environment (Fig. 17).

3. We have conducted additional quantitative comparative evaluations of the goal-tracking performance across 200 simulation trials (Sec. A.5, Fig. 11). In our overall comparative results, we have also added an additional baseline - ASE using a latent difference $\mathcal{L}=-(\phi(s’)-\phi(s))^Tz)$.

4. We have added a comparison with two reward-dense methods (one hierarchical and one standard RL controller), showing that our approach is competitive in terms of goal-tracking performance even in zero-shot.

5. We have rewritten parts of the Methodology and Relevant work to be more accessible to readers unfamiliar with skill discovery terminology. We have added a Preliminaries section (Sec 2.1) to describe our MDP formulation and all relevant notation.

6. We added several relevant works on empowerment, options learning and goal-conditioned RL in the Related work (Sec. 2.2).

7. We included a reproducibility statement.

---

### Author Response · Authors · 2024-12-04
**Brief summary of improvements**

We thank the Reviewers and Area Chairs for their feedback and discussions, which helped significantly improve our manuscript. We have addressed all of the concerns raised by the Reviewers, which we summarise briefly below.

We have made our contributions clearer by highlighting how **our approach uses the full Mutual Information (MI) objective** rather than a simplified version like prior work, and how this **ensures a wider range of skills** can be learned. We demonstrated that our method is **competitive with dense reward baselines on goal-tracking quadrupedal tasks**, even in zero-shot. We have further shown that our objective can **work well with non-Euclidean distance metrics, too** - such as temporal distance, highlighting its generality. In addition, we demonstrate that our method **can learn meaningful skills in more complex maze environments**, and outperform prior work at navigating them in zero-shot. Finally, we have rewritten parts of the methodology and relevant work to **make the language and notation clearer and more accessible**.

We hope that these changes and additions address all comments and suggestions by the Reviewers, and have significantly improved our manuscript.

---

### Meta-Review · Area_Chair_ER79 · 2024-12-21

**Metareview:**

The AC read the paper, reviewers and discussions. As suggested by the general opinions from reviewers, the quality of this paper is below the acceptance threshold.
The paper investigates unsupervised RL for constrained skill discovery, with applications to quadrupedal robot locomotion control. It incrementally improves results based on previous works e.g. Lipschitz-constrained Skill Discovery (LSD). It takes advantage of mutual information between skills and states. A norm-matching objective is proposed to achieve broader state space.
As mentioned by reviewers: previous methods such as LSD conducted comparative experiments on multiple tasks in simulation. While this paper only report experiments on velocity tracking on the ANYmal quadruped robot, falling in the gulf between algorithmic and experimental papers.

**Additional Comments On Reviewer Discussion:**

The reviewers kept the general evaluations unchanged.

---

### Decision · Program_Chairs · 2025-01-22

Reject